# Sample Efficient Deep Reinforcement Learning via Uncertainty Estimation

**Vincent Mai, Kaustubh Mani and Liam Paull** [*]
Robotics and Embodied AI Lab
Mila - Quebec Institute of Artificial Intelligence
Université de Montréal, Quebec, Canada
`{vincent.mai,kaustubh.mani,liam.paull}@umontreal.ca`

## Abstract

In model-free deep reinforcement learning (RL) algorithms, using noisy value estimates to supervise policy evaluation and optimization is detrimental to the sample efficiency. As this noise is heteroscedastic, its effects can be mitigated using uncertainty-based weights in the optimization process. Previous methods rely on sampled ensembles, which do not capture all aspects of uncertainty. We provide a systematic analysis of the sources of uncertainty in the noisy supervision that occurs in RL, and introduce inverse-variance RL, a Bayesian framework which combines probabilistic ensembles and Batch Inverse Variance weighting. We propose a method whereby two complementary uncertainty estimation methods account for both the Q-value and the environment stochasticity to better mitigate the negative impacts of noisy supervision. Our results show significant improvement in terms of sample efficiency on discrete and continuous control tasks.

## 1 Introduction

Deep reinforcement learning (DRL) methods have proven to be powerful at solving sequential decision-making tasks across domains (Silver et al., 2016; OpenAI et al., 2019). Combining the flexibility of the reinforcement learning framework with the representational power of deep neural networks enables policy optimization in complex and high-dimensional environments with unknown dynamics models to maximize the expected cumulative reward (Sutton & Barto, 2018).

An important limitation of DRL methods is their sample inefficiency: an enormous amount of data is necessary and makes training expensive. This makes applying DRL in the real world challenging, for example in robotics (Sünderhauf et al., 2018; Dulac-Arnold et al., 2019). In tasks like manipulation, sample collection is a slow and costly process (Liu et al., 2021). It is even more expensive in risk-averse applications like autonomous driving (Kothari et al., 2021).

Among the current state-of-the-art approaches to improve learning efficiency, a promising direction is to exploit the prevalence of uncertainty in the underlying DRL algorithm. By adopting a Bayesian framework, we can consider the sampled quantities in DRL as random variables and leverage information about their distributions to improve the learning process (Osband et al., 2018). In this paper, we consider the particular problem of unreliable supervision in the temporal difference update and the policy optimization process. In DRL, value predictions are used to supervise the training: in temporal difference-based algorithms, they are included in bootstrapped target values which are used as labels; in actor-critic frameworks, the policy is trained to optimize them. That these value predictions are noisy slows the learning and brings instability (Kumar et al., 2019; 2020). The amount of noise in the supervision depends on the uncertainty of the value prediction, which evolves during the training process and depends on the state (and action) evaluated. It is therefore *heteroscedastic*.

While there is an extensive body of literature focused on using the uncertainty of the value prediction to guide the exploration/exploitation trade-off (Dearden et al., 1998; Strens, 2001; Osband et al., 2016; Pathak et al., 2017; Chen et al., 2017; Osband et al., 2018; Fortunato et al., 2019; Osband et al., 2019; Flennerhag et al., 2020; Clements et al., 2020; Jain et al., 2021; Aravindan & Lee, 2021), there are very few works focused on leveraging it to mitigate the impact of unreliable supervision.

---

[*]Canada CIFAR AI Chair

Distributional RL (Bellemare et al., 2017) considers the value function as a distribution to be learned as such. It is orthogonal to our proposition: we consider the uncertainty of the labels used to learn a scalar value function. In the offline RL setting, where the dataset is limited, uncertainty-weighted actor-critic (UWAC) (Wu et al., 2021) uses inverse-variance weighting to discard out-of-distribution state-action pairs using Monte Carlo dropout (Gal & Ghahramani, 2016) for uncertainty estimation.

Closer to our work, Lee et al. (2021) propose SUNRISE, in which each sample of the Bellman backup in the TD update step is weighted to lower the importance of the targets which have a high standard deviation. The weights $w(s', a')$ are computed based on a sigmoïd of the negative standard deviation $\hat{Q}_{\text{std}}(s', a')$ scaled by a temperature hyperparameter $T$, and then offset such that they are between 0.5 and 1: $w(s, a) = \sigma(-\hat{Q}_{\text{std}}(s', a') * T) + 0.5$. The uncertainty of the target is estimated by sampled ensembles. While SUNRISE proposes other contributions such as an exploration bonus, the heuristic weighting scheme and the limitations of sampled ensembles in capturing the predictive uncertainty leave space for improvement in the mitigation of the effects of unreliable supervision.

We propose inverse-variance reinforcement learning (IV-RL). IV-RL also uses weights to reduce the importance of uncertain targets in training. It does so by addressing the problem from two viewpoints. First, we use variance networks (Kendall & Gal, 2017), whose loss function for regression is the negative log-likelihood instead of the L2 distance. For a given state-action pair $(s, a)$, the network learns the target's noise, due for example to the stochasticity of the environment or the update of the policy. It then naturally down-weights the highly noisy samples in the training process. Second, we use variance ensembles (Lakshminarayanan et al., 2017) to estimate the uncertainty of the target due to the prediction of $Q(s', a')$ during the temporal-difference update. We merge the predicted variances of several variance networks through a mixture of Gaussians, which has been shown to be a reliable method to capture predictive uncertainty (Ovadia et al., 2019). We then use Batch Inverse-Variance (BIV) (Mai et al., 2021), which has been shown to significantly improve the performance of supervised learning with neural networks in the case of heteroscedastic regression. BIV is normalized, which makes it ideal to cope with different and time-varying scales of variance. We show analytically that these two different variance predictions for the target are complementary and their combination leads to consistent and significant improvements in the sample efficiency and overall performance of the learning process.

In summary, our contribution is threefold:

1. We present a systematic analysis of the sources of uncertainty in the supervision of model-free DRL algorithms. We show that the variance of the supervision noise can be estimated with two complementary methods: negative log-likelihood and variance ensembles.

2. We introduce IV-RL, a framework that accounts for the uncertainty of the supervisory signal by weighting the samples in a mini-batch during the agent's training. IV-RL uses BIV, a weighting scheme that is robust to poorly calibrated variance estimation.[1]

3. Our experiments show that IV-RL can lead to significant improvements in sample efficiency when applied to Deep Q-Networks (DQN) (Mnih et al., 2013) and Soft-Actor Critic (SAC) (Haarnoja et al., 2018).

In section 2, we introduce BIV as a weighting scheme for heteroscedastic regression, and variance ensembles as an uncertainty estimation method. We analyse the sources of uncertainty in the target in section 3, where we also introduce our IV-RL framework. We finally present our experimental results in section 4.

## 2 BACKGROUND AND PRELIMINARIES

### 2.1 BATCH INVERSE-VARIANCE WEIGHTING

In supervised learning with deep neural networks, it is assumed that the training dataset consists of inputs $x_k$ and labels $y_k$. However, depending on the label generation process, the label may be noisy. In regression, we can model the noise as a normal distribution around the true label: $\tilde{y}_k = y_k + \delta_k$ with $\delta_k \sim \mathcal{N}(0, \sigma_k^2)$. In some cases, the label generation process leads to different variances for the label noises. When these variances can be estimated, each sample is a triplet $(x_k, \tilde{y}_k, \sigma_k^2)$.

---

[1]The code for IV-RL is available at `https://github.com/montrealrobotics/iv_rl`.

Batch Inverse-Variance (BIV) weighting (Mai et al., 2021) leverages the additional information $\sigma_k^2$, which is assumed to be provided, to learn faster and obtain better performance in the case of heteroscedastic noise on the labels. Applied to L2 loss, it optimizes the neural network parameters $\theta$ using the following loss function for a mini-batch $D$ of size $K$ [2]:

$$\mathcal{L}_{\text{BIV}}(D, \theta) = \left( \sum_{k=0}^{K} \frac{1}{\sigma_k^2 + \xi} \right)^{-1} \sum_{k=0}^{K} \frac{(f_\theta(x_k) - \tilde{y}_k)^2}{\sigma_k^2 + \xi} \tag{1}$$

This is a normalized weighted sum with weights $w_k = 1/(\sigma_k^2 + \xi)$. Normalizing in the mini-batch enables control of the effective learning rate, especially in cases where the training data changes over time, such as in DRL. By focusing on the relative scale of the variances instead of their absolute value, it also provides robustness to poor scale-calibration of the variance estimates.

As explained in Mai et al. (2021), $\xi$ is a hyperparameter that is important for the stability of the optimization process. A higher $\xi$ limits the highest weights, thus preventing very small variance samples from dominating the loss function for a mini-batch. However, by controlling the discrimination between the samples, $\xi$ is also key when the variance estimation is not completely trusted. It provides control of the effective mini-batch size $EBS$, according to:

$$EBS = \frac{\left( \sum_k^K w_k \right)^2}{\sum_k^K w_k^2} = \frac{\left( \sum_k^K \frac{1}{(\sigma_k^2 + \xi)} \right)^2}{\sum_k^K \frac{1}{(\sigma_k^2 + \xi)^2}} \tag{2}$$

For example, imagine a mini-batch where most samples have very high variances, and only one has a very low variance. If $\xi = 0$, this one low-variance sample is effectively the only one to count in the mini-batch, and $EBS$ tends towards 1. Increasing $\xi$ would give more relative importance to the other samples, thus increasing $EBS$. With a very high $\xi$ compared to the variances, all weights are equal, and $EBS$ tends towards $K$; in this case, the BIV loss tends towards $L2$.

**Tuning the $\xi$ parameter**  The simplest way to set $\xi$ is to choose a constant value as an additional hyperparameter. However, the best value is difficult to evaluate a priori and can change when the profile of variances changes during a task, as is the case in DRL.

It is instead possible to numerically compute the value of $\xi$ which ensures a minimal $EBS$ for each mini-batch. This method allows $\xi$ to automatically adapt to the different scales of variance while ensuring a minimal amount of information from the dataset is accounted for by the algorithm. The minimal $EBS$ is also a hyper-parameter, but it is easier to set and to transfer among environments, as it is simply a fraction of the original batch size. As such, it can be set as a batch size ratio.

## 2.2  ESTIMATING THE UNCERTAINTY OF A NEURAL NETWORK PREDICTION

The predictive uncertainty of a neural network can be considered as the combination of aleatoric and epistemic uncertainties (Kendall & Gal, 2017). Aleatoric uncertainty is irreducible and characterizes the non-deterministic relationship between the input and the desired output. Epistemic uncertainty is instead related to the trained model: it depends on the information available in the training data, the model's capacity to retain it, and the learning algorithm (Hüllermeier & Waegeman, 2021). There is currently no principled way to quantify the amount of task-related information present in the input, the training data, or the model. The state of the art for predictive uncertainty estimation instead relies on different sorts of proxies. These sometimes capture other elements, such as the noise of the labels, which we can use to our advantage. We focus here on the relevant methods to our work.

### 2.2.1  SAMPLED ENSEMBLES

Several networks independently train an ensemble of size $N$ that can be interpreted as a distribution over predictions. The expected behaviour is that different networks will only make similar predictions if they were sufficiently trained for a given input. The sampled variance of the networks' outputs is thus interpreted as epistemic uncertainty. It is possible to include a random Bernoulli mask

---

[2]We replaced $\epsilon$ in Mai et al. (2021) with $\xi$ to avoid confusion with the reinforcement learning conventions.

of probability $p$ to each training sample, to ensure that each network undergoes different training. This method, used by Clements et al. (2020) and Lee et al. (2021), has the same principle as single network Monte-Carlo dropout (Gal & Ghahramani, 2016). As the variance is sampled, the standard deviation is usually on the same scale as the prediction.

When used at the very beginning of the training process, sampled ensembles present one particular challenge: as the networks are initialized, they all predict small values. The initial variance, instead of capturing the lack of knowledge, is then underestimated. To address this problem, Randomized Prior Functions (RPFs) enforce a prior in the variance by pairing each network with a fixed, untrained network which adds its predictions to the output (Osband et al., 2019). RPFs ensure a high variance at regions of the input space which are not well explored, and a lower variance when the trained networks have learned to compensate for their respective prior and converge to the same output. The scale of the prior is a hyper-parameter.[3]

### 2.2.2 VARIANCE NETWORKS

With variance networks, the uncertainty is predicted using loss attenuation (Nix & Weigend, 1994; Kendall & Gal, 2017). A network outputs two values in its final layer given an input $x$: the predicted mean $\mu(x)$ and variance $\sigma^2(x)$. Over a minibatch $D$ of size $K$, the network parameters $\theta$ are optimized by minimizing the negative log-likelihood of a heteroscedastic Gaussian distribution:

$$\mathcal{L}_{LA}(D, \theta) = \sum_{k=0}^{K} \frac{1}{K} \frac{(\mu_\theta(x_k) - y(x_k))^2}{\sigma_\theta^2(x_k)} + \ln \sigma_\theta^2(x_k) \tag{3}$$

Variance networks naturally down-weight the labels with high variance in the optimization process. The variance prediction is trained from the error between $\mu_\theta(x)$ and $y(x)$. Hence, noise on the labels or changes in the regression task over time will also be captured by loss attenuation. As the variance is predicted by a neural network, it may not be well-calibrated and may be overestimated (Kuleshov et al., 2018; Levi et al., 2020; Bhatt et al., 2021). This can (1) give wrong variance estimates but also (2) affect the learning process by ignoring a sample if the variance estimate is too high.

### 2.2.3 VARIANCE ENSEMBLES

Variance ensembles, or deep ensembles (Lakshminarayanan et al., 2017), are ensembles composed of variance networks. The predictive variance is given by a Gaussian mixture over the variance predictions of each network in the ensemble. This method, when trained, is able to capture uncertainty more reliably than others (Ovadia et al., 2019), with $N = 5$ networks in the ensemble being sufficient. Similarly to sampled ensembles, variance ensembles suffer from underestimated early epistemic variance estimation. However, compared to variance networks, they seem empirically less prone to calibration issues (1) in the final variance estimation because the mixture of Gaussians dampens single very high variances, and (2) in the learning process because even if one network does not learn correctly, the others will.

For better readability, we will use the terms var-network and var-ensemble in the rest of the paper.

### 2.3 UNCERTAINTY AND EXPLORATION IN DRL

While our work focuses on using uncertainty estimates to mitigate the impact of unreliable supervision, we can take advantage of the structure in place to better drive the exploration/exploitation trade-off. In particular, we used BootstrapDQN (Osband et al., 2016) for exploration. In this method, a single network is sampled from an ensemble at the beginning of each episode to select the action. This method is improved with the previously described RPFs (Osband et al., 2018). In continuous settings, we instead followed Lee et al. (2021) and added an Upper Confidence Bound (UCB) exploration bonus based on uncertainty prediction. As the variance in UCB is added to $Q$-values, it must be calibrated: we evaluate it with sampled ensembles. Possible interactions between exploration strategies and IV-RL are discussed in Appendix H.

---

[3]Using RPFs in the context of DRL can destabilize the exploration due to a significant initial bias added to the value of each state-action. It must be compensated for with uncertainty-aware exploration, such as Bootstrap (Osband et al., 2016), where it leads to a significant improvement in performance for exploration-based tasks.

# 3 INVERSE-VARIANCE REINFORCEMENT LEARNING

## 3.1 TARGET UNCERTAINTY IN REINFORCEMENT LEARNING

Many model-free DRL algorithms use temporal difference updates. In methods such as DQN (Mnih et al., 2013), PPO (Schulman et al., 2017) and SAC (Haarnoja et al., 2018), a neural network is trained to predict the $Q$-value of a given state-action pair $Q^\pi(s, a)$ [4] by minimizing the error between the target $T(s, a)$ and its prediction $\hat{Q}(s, a)$. $T(s, a)$ is computed according to Bellman's equation:

$$T(s, a) = r + \gamma \bar{Q}(s', a') \tag{4}$$

$s'$ and $r$ are sampled from the environment given $(s, a)$, and $a'$ is sampled from the current policy given $s'$. $\bar{Q}(s', a')$ is predicted by a copy of the $Q$-network (called the target network) which is updated less frequently to ensure training stability. The neural network's parameters $\theta$ are optimized using stochastic gradient descent to minimize the following $L2$ loss function:

$$\mathcal{L}_\theta = \left\| T(s, a) - \hat{Q}_\theta(s, a) \right\|^2 \tag{5}$$

### 3.1.1 THE TARGET AS A RANDOM VARIABLE

The target $T(s, a)$ is a noisy approximation of $Q^\pi(s, a)$ that is distributed according to its distribution $p_T(T|s, a)$. The generative model used to produce samples of $T(s, a)$ is shown in Figure 1, and has the following components:

1. if the reward $r$ is stochastic, it is sampled from $p_R(r|s, a)$ [5];
2. if the environment dynamics are stochastic, the next state $s'$ is sampled from $p_{S'}(s'|s, a)$;
3. if the policy is stochastic $a'$ is sampled from the policy $\pi(a'|s')$;
4. $\bar{Q}$ is a prediction from a var-network $p_{\bar{Q}}(\bar{Q}|s', a')$;
5. $T$ is deterministically generated from $r$ and $\bar{Q}$ by equation (4).

As the variance of the noise of $T(s, a)$ is not constant, training the $Q$-network using $\mathcal{L}_\theta$ as in equation (5) is regression on heteroscedastic noisy labels.

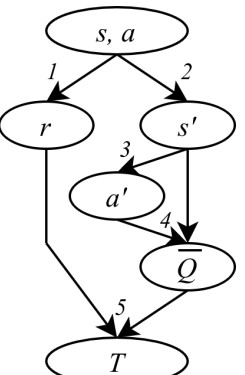

Figure 1: Bayesian network representing the target sampling process

### 3.1.2 VARIANCE OF THE TARGET

As seen in section 2.1, BIV can be used to reduce the impact of heteroscedastic noisy labels in regression, provided estimates of the label variances. We thus aim to evaluate $\sigma_T^2(T|s, a)$ based on the sampling process described in section 3.1.1. $r$ and $Q$-value estimation are independent given $s$ and $a$, hence:

$$\sigma_T^2(T|s, a) = \sigma_R^2(r|s, a) + \gamma^2 \sigma_{S'A'\bar{Q}}^2(\bar{Q}|s, a) \tag{6}$$

where $p_{S'A'\bar{Q}}$ is the compound probability distribution based on components 2-4 in Figure 1:

$$p_{S'A'\bar{Q}}(\bar{Q}|s, a) = \iint p_{\bar{Q}}(\bar{Q}|s', a')p_{S'A'}(s', a'|s, a)da'ds' \tag{7}$$

where $p_{S'A'}(s', a'|s, a) = p_{A'}(a'|s')p_{S'}(s'|s, a)$. Using the law of total variance, the variance of $\bar{Q}$ is given by:

$$\sigma_{S'A'\bar{Q}}^2(\bar{Q}|s, a) = \mathbb{E}_{S'A'}\left[\sigma_{\bar{Q}}^2(\bar{Q}|s', a')|s, a\right] + \sigma_{S'A'}^2(\mathbb{E}_{\bar{Q}}[\bar{Q}|s', a']|s, a) \tag{8}$$

Plugging (8) into (6) gives:

$$\sigma_T^2(T|s, a) = \gamma^2 \underbrace{\left(\mathbb{E}_{S'A'}\left[\sigma_{\bar{Q}}^2(\bar{Q}|s', a')|s, a\right]\right)}_{\text{Predictive variance of Q-network}} + \underbrace{\gamma^2\left(\sigma_{S'A'}^2(\mathbb{E}_{\bar{Q}}[\bar{Q}|s', a']|s, a)\right) + \sigma_R^2(r|s, a)}_{\text{Policy and environment induced variance}}$$

$$\tag{9}$$

---

[4]In this paper, we focus on $Q$-values, although the principle can also apply to state values.

[5]Stochastic reward can happen for example if the reward is corrupted or estimated (Romoff et al., 2018; Campbell et al., 2015)

We can identify two distinct components in equation 9 that contribute to the overall variance of the target. The first is the (expectation of the) variance that is due to the uncertainty in the neural network prediction of the value function, $\mathbb{E}_{S'A'}[\sigma_{\bar{Q}}^2 (\bar{Q}|s', a')]$. The second is the uncertainty due to the stochasticity of the environment and of the policy, $\sigma_R^2(r|s, a) + \gamma^2 \sigma_{S'A'}^2 (\mathbb{E}_{\bar{Q}} [\bar{Q}|s', a'])$.

**Uncertainty in neural network prediction**    For a given policy $\pi$ and a given $s', a'$, the agent may not have seen enough samples to have an accurate approximation of $Q^{\pi}(s', a')$. This corresponds to an epistemic source of uncertainty that should be captured by sampled ensembles. However, as $\pi$ is updated, the regression target, $Q^{\pi}(s', a')$, is also changing. This can be interpreted as variability in the underlying process which will be captured by var-networks. We can thus combine both sampling-based and var-network methods and evaluate $\sigma_{\bar{Q}}^2 (\bar{Q}(s', a'))$ with var-ensembles.

We assume that the estimate of $\sigma_{\bar{Q}}^2(\bar{Q}|s', a')$ given a sampled $(s', a')$ is unbiased and can therefore use it to directly approximate the expectation $\mathbb{E}_{S'A'}[\sigma_{\bar{Q}}^2 (\bar{Q}|s', a')]$. These values are used in the BIV loss $\mathcal{L}_{BIV}$ (equation 1) across a mini-batch sampled from the replay buffer. In this case, $\xi$ is used to control the trust in the variance estimation, as explained in section 2.1.

**Stochastic environment and policy**    The other potential source of variance in the target is the result of the stochasticity of the environment encapsulated by $p_R (r|s, a)$ and $p_{S'}(s'|s, a)$ and of the policy represented by $\pi(a'|s')$. Note that in model-free RL, we have no explicit representation of $p_R (r|s, a)$ or $p_{S'}(s'|s, a)$, which are necessary to estimate this source of uncertainty.

$Q^{\pi}(s, a)$ is defined as the expected value of the return. As a result, even in the case where $\bar{Q}(s', a')$ = $Q^{\pi}(s, a)$ in (4), there is still noise over the value of the *target* that is being used as the label due to the stochasticity of the environment and policy that generate $r$, $a'$ and $s'$. Assuming a zero mean and normally distributed noise, this underlying stochasticity of the generating process is well-captured by a var-network with a loss attenuation using the negative log-likelihood formulation described in equation (3).

### 3.1.3 LOSS FUNCTION FOR IV-RL

Based on the motivation above, we propose our IV-RL loss over minibatch $D$ of size $K$ as a linear combination of $\mathcal{L}_{\text{BIV}}$ and $\mathcal{L}_{\text{LA}}$, balanced by hyperparameter $\lambda$:

$$\mathcal{L}_{\text{IVRL}}(D, \theta) = \mathcal{L}_{\text{BIV}}(D, \theta) + \lambda \mathcal{L}_{\text{LA}}(D, \theta)$$

$$= \sum_{k=0}^{K} \left[ \left( \sum_{j=0}^{K} \frac{1}{\gamma^2 \sigma_{\bar{Q},j}^2 + \xi} \right)^{-1} \frac{\left( \mu_{\hat{Q}_\theta}(s_k, a_k) - T(s_k, a_k) \right)^2}{\gamma^2 \sigma_{\bar{Q},k}^2 + \xi} \right.$$

$$\left. + \frac{\lambda}{K} \left( \frac{\left( \mu_{\hat{Q}_\theta}(s_k, a_k) - T(s_k, a_k) \right)^2}{\sigma_{\hat{Q}_\theta}^2 (s_k, a_k)} + \ln \sigma_{\hat{Q}_\theta}^2 (s_k, a_k) \right) \right] \qquad (10)$$

The var-network with parameters $\theta$ predicts both $\mu_{\hat{Q}_\theta}(s_k, a_k)$ and $\sigma_{\hat{Q}_\theta}^2 (s_k, a_k)$ for each element $k$ of the mini-batch. $\sigma_{\bar{Q},k}^2$ is instead provided by the target var-ensemble which predicts $\bar{Q}(s_k', a_k')$ when estimating the target $T(s_k, a_k)$. In $\mathcal{L}_{\text{BIV}}$ of equation (1), $\sigma_k^2$ is replaced by $\gamma^2 \sigma_{\bar{Q},k}^2$ according to equation (9). Empirically, the value of $\lambda$ is usually optimal around 5 or 10. Its impact is studied more in details in appendix C. High-variance samples generated from the target estimation will be down-weighted in the BIV loss, while high-variance samples due to the stochasticity of the underlying environment and policy will be down-weighted in the LA loss. In the remainder of the paper, we show how this loss can be applied to different architectures and algorithms, and demonstrate that in many cases it significantly improves sample efficiency.

### 3.2 Q-VALUE UNCERTAINTY AND ACTOR-CRITIC STRUCTURES

In section 3.1, we discussed how the target's uncertainty can be quantified and how the Bellman update can then be interpreted as a heteroscedastic regression problem. This is applicable in most

model-free DRL algorithms, whether they are based on $Q$-learning or policy optimization. In the special case of actor-critic algorithms, the state-values or $Q$-values predicted by the critic network are also used to train the policy $\pi_\phi$'s parameters by gradient ascent optimization. An estimate of their variance can also be used to improve the learning process.

The objective is to maximize the expected Q-value:

$$\mathbb{E}_{s \sim D, a \sim \pi_\phi(s)}[Q(s, a)] \tag{11}$$

where $D$ is the state distribution over the probable agent trajectories. The expectation is computed by sampling a mini-batch of size $K$ from a replay buffer containing state-action pairs $(s_k, a_k)$. The $Q$-value is approximated by the critic as $\hat{Q}(s_k, a_k)$. The actor's parameters $\phi$ are then trained to maximize the unweighted average:

$$1/K \sum_{k=0}^{K} \hat{Q}(s_k, a_k) \tag{12}$$

If we instead consider the critic's estimation $\hat{Q}(s_k, a_k)$ as a random variable sampled from $p_{\hat{Q}}(\hat{Q}|s, a)$, with mean $\mu_{\hat{Q}}(s_k, a_k)$ and variance $\sigma_{\hat{Q}}^2(s_k, a_k)$, we can instead infer the expected value in equation (11) using Bayesian estimation (Murphy, 2012):

$$\left( \sum_{k=0}^{K} 1/\sigma_{\hat{Q}}^2(s_k, a_k) \right)^{-1} \sum_{k=0}^{K} \frac{1}{\sigma_{\hat{Q}}^2(s_k, a_k)} \mu_{\hat{Q}}(s_k, a_k) \tag{13}$$

The normalized inverse-variance weights are a direct fit with the BIV loss in equation 1: we therefore also use BIV to train the actor. As the target and the $Q$-networks have the same structure, $\sigma_{\hat{Q}}^2(s_k, a_k)$ can be estimated using the same var-ensembles used in section 3.1.2.

## 3.3 Algorithms

We have adapted IV-RL to DQN and SAC to produce IV-DQN and IV-SAC. Comprehensive pseudo-code, as well as implementation details for each algorithm, can be found in appendix A.

## 4 Results

We have tested IV-RL across a variety of different environments, using IV-DQN for discrete tasks and IV-SAC for continuous tasks. To determine the effects of IV-RL, we compare its performance to the original DQN and SAC, but also the ensemble-based baselines upon which IV-DQN and IV-SAC were built: BootstrapDQN and EnsembleSAC. We also include SUNRISE (Lee et al., 2021).

All ensemble-based methods use an ensemble size of $N = 5$. Unless specified otherwise, each result is the average of runs over 25 seeds: 5 for the environment $\times$ 5 for network initialization. The hyperparameters are the result of a thorough fine-tuning process explained in appendix B. As it uses ensembles, IV-RL is computationally expensive: computation time is discussed in Appendix D.

## 4.1 IV-DQN

We tested IV-DQN on discrete control environments selected to present different characteristics. From OpenAI Gym (Brockman et al., 2016), LunarLander is a sparse reward control environment and MountainCar is a sparse reward exploration environment. From BSuite[6] (Osband et al., 2020), Cartpole-Noise is a dense reward control environment. The BootstrapDQN baseline follows Osband et al. (2018) as explained in section 2.3. The reported SUNRISE results are obtained by applying the SUNRISE weights to BootstrapDQN. The learning curves are shown in figure 2, where the results are averaged on a 100 episode window.

IV-DQN significantly outperforms BootstrapDQN and SunriseDQN on control-based environments. However, in the exploration-based Mountain-Car where the challenge is to find the state that produces some reward, there is very little additional information in every other state for IV-DQN to improve over BootstrapDQN. All weighting schemes thus perform comparably.

---

[6]The results from BSuite are averaged over 20 runs with different environment and network seeds

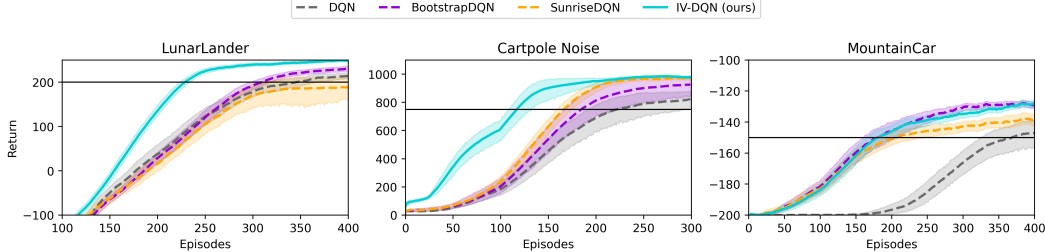

Figure 2: Using IV-RL shows improved performance over baseline methods in Cartpole Noise and LunarLander, and does not impact MountainCar.

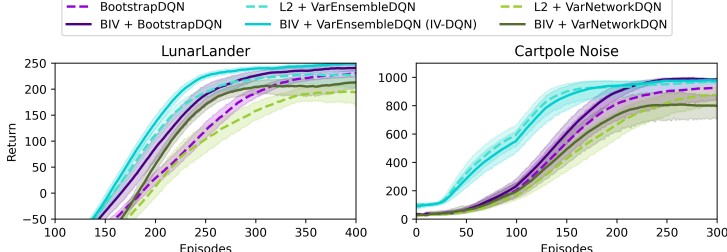

Figure 3: Ablation study: depending on the environment, the BIV or the var-ensemble component is the most important factor of improvement.

Table 1 shows the median number of episodes necessary to reach a given score for which the environment is considered as solved. IV-DQN shows a clear improvement over baselines in sample efficiency in both control-based environments but fails to improve the exploration in Mountain Car.

Table 1:  25th - 50th - 75th percentiles of the number of episodes necessary for the return averaged with a 100-episode window to reach the solved score on different environments. IV-DQN shows significant improvements in sample efficiency when the environment is not exploration-based.

|  | LunarLander (200) | Cartpole-Noise (750) | MountainCar (-150) |
|---|---|---|---|
| DQN | 296 - 316 - 349 | 171 - 193 - max | 304 - 333 - 403 |
| BootstrapDQN | 287 - 305 - 317 | 160 - 174 - 196 | **134 - 149 - 206** |
| SunriseDQN | 291 - 309 - 368 | 155 - 165 - 175 | 152 - 197 - 257 |
| IV-DQN (ours) | **220 - 227 - 239** | **105 - 112 - 117** | 142 - 163 - 200 |

In Figure 3 we perform an ablation study for LunarLander and Cartpole-Noise. In L2 + VarEnsembleDQN, the loss function $\mathcal{L}_{\text{BIV}}$ in equation 10 is replaced by the $L2$ loss. In BIV + BootstrapDQN, sampled ensembles are used instead of var-ensembles, and trained only with $\mathcal{L}_{\text{BIV}}$. More details about these ablations are found in appendix A. In both cases, the use of BIV or var-ensembles improves the performance, and their combination leads to the best sample efficiency. In Cartpole-Noise, var-ensembles carry most of the improvement, maybe because the low amount of states leads to low epistemic uncertainty. We also show that using one single var-network as opposed to var-ensembles is sub-optimal. This is likely due to the unstable nature of the single uncertainty estimate, which affects the learning process. More details are shown in appendix E and G. The use of var-ensembles stabilizes the variance estimation, as explained in section 2.2.3.

## 4.2 IV-SAC

IV-SAC was applied to different continuous control environments from OpenAI Gym (Brockman et al., 2016) as implemented by MBBL (Wang et al., 2019). EnsembleSAC is a baseline using ensembles with an UCB exploration bonus (Lee et al., 2021). Table 2 shows the average return after 100k and 200k steps on 25 seeds. We note that the addition of an ensemble instead of a single network, along with the uncertainty-based exploration bonus, already allows the performance to increase compared to SAC. Except in Ant, the SUNRISE weights do not seem to lead to consistently better results. In comparison, IV-SAC leads to significant improvements in performance.

This improvement in performance after a fixed amount of training steps can be explained by an improved sample efficiency. This can be seen in figure 4. Even when IV-SAC's return is not signifi-

Table 2: Performance at 100K and 200K time steps (100 and 200 episodes) for several robotics environments in OpenAI Gym. The results show the mean and standard error over 25 runs.

| | | Walker | HalfCheetah | Hopper | Ant | SlimHuman. |
|---|---|---|---|---|---|---|
| 100k steps | SAC | $-392 \pm 187$ | $3211 \pm 136$ | $322 \pm 177$ | $724 \pm 29$ | $815 \pm 74$ |
| | EnsembleSAC | $-389 \pm 209$ | $3938 \pm 112$ | $1597 \pm 152$ | $852 \pm 27$ | $1043 \pm 31$ |
| | SunriseSAC | $46 \pm 213$ | $3879 \pm 204$ | $1618 \pm 195$ | $834 \pm 130$ | $1077 \pm 41$ |
| | IV-SAC (ours) | $\mathbf{857 \pm 231}$ | $\mathbf{4260 \pm 118}$ | $\mathbf{2237 \pm 116}$ | $\mathbf{948 \pm 46}$ | $\mathbf{1122 \pm 34}$ |
| 200k steps | SAC | $371 \pm 189$ | $3978 \pm 148$ | $1635 \pm 162$ | $985 \pm 82$ | $1020 \pm 52$ |
| | EnsembleSAC | $1337 \pm 278$ | $4757 \pm 92$ | $2652 \pm 91$ | $981 \pm 65$ | $1294 \pm 39$ |
| | SunriseSAC | $1423 \pm 295$ | $4785 \pm 228$ | $2572 \pm 119$ | $\mathbf{1462 \pm 154}$ | $1383 \pm 56$ |
| | IV-SAC (ours) | $\mathbf{3009 \pm 193}$ | $\mathbf{5451 \pm 151}$ | $\mathbf{2889 \pm 50}$ | $1281 \pm 101$ | $\mathbf{2023 \pm 114}$ |

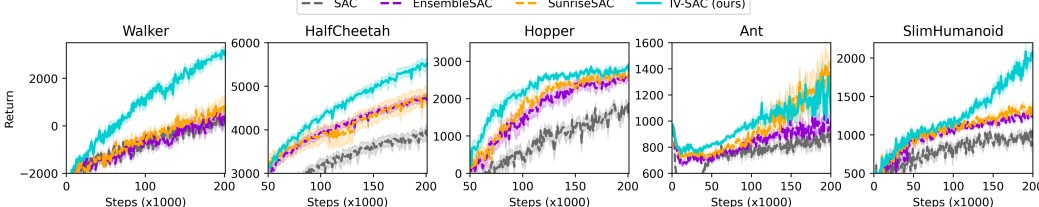

Figure 4: IV-SAC learns faster and leads to significantly better results than the baselines.

cantly better than the baselines at 200k steps, such as in Ant or Hopper, it clearly is learning faster, which is also reflected by the scores at 100k steps.

Similarly to IV-DQN, we can separate the contribution from BIV and loss attenuation, as shown in the two first plots of figure 5. While both BIV and var-ensembles alone have a significant impact in Walker, it's only their combination that brings the most improvement in HalfCheetah. Similarly to the discrete control case, a simple var-network leads to a small improvement over SAC, which is discussed in appendix G. We also show in figure 5 that the BIV weights provided with var-ensemble variance estimations than the SUNRISE weights, or even the inverse variance weights of UWAC (Wu et al., 2021). The normalization of BIV allows it to better cope with changes in the variance predictions, as seem in appendices E and F.

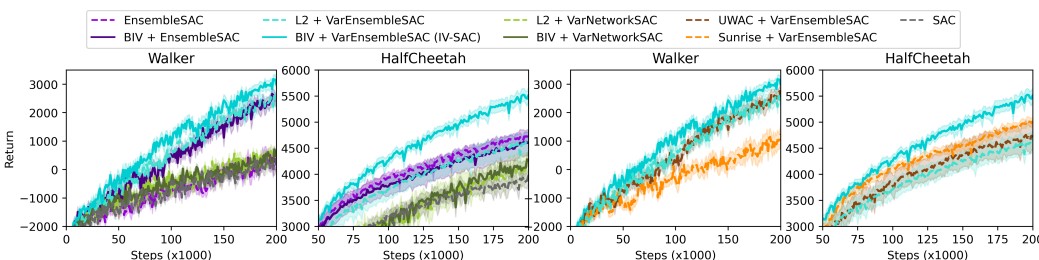

Figure 5: **Ablation Study:** (first two figures) Impact of using different uncertainty estimation methods (last two figures) Comparing different weighting schemes with var-ensembles.

## 5 CONCLUSION

We present Inverse Variance Reinforcement Learning (IV-RL), a framework for model-free deep reinforcement learning which leverages uncertainty estimation to enhance sample efficiency and performance. A thorough analysis of the sources of noise that contribute to errors in the target motivates the use of a combination of Batch Inverse Variance (BIV) weighting and variance ensembles to estimate the variance of the target and down weight the uncertain samples in two complementary ways. Our results show that these two components are beneficial and that their combination significantly improves the state of the art in terms of learning efficiency, in particular for control tasks. We applied IV-RL to DQN and SAC, but it can be adapted to any model-free algorithm. Model-based, active or continuous learning could also benefit from the underlying ideas presented in this paper. We thus believe that IV-RL is a significant step to enable the application of DRL to real-world problems such as robotics, where sample efficiency is the main challenge.

## ETHICS STATEMENT

In this work, we present a method to enhance the sample efficiency of deep reinforcement learning algorithms. As such, it is agnostic to the applications, and per se does not raise any particular ethical issue. We however strongly encourage the user of our algorithm to ensure they have carefully thought about the ethical issues related to the particular field of application, such as medicine, robotics, communication, finance, etc.

As environmental sustainability can also be considered an ethical issue (Université de Montréal, 2018), we publish the carbon footprint of our work.

Experiments were conducted using a private infrastructure, which has a carbon efficiency of 0.028 $kgCO_2eq/kWh$. A cumulative of 12367 days, or 296808 hours, of computation was mainly performed on the hardware of type RTX 8000 (TDP of 260W). We assume full power usage of the GPUs, although this was not always the case.

Total emissions are estimated to be 2160.76 $kgCO_2eq$ of which 0 percents were directly offset. This is equivalent to 8730 km driven by an average car, or 1.08 metric tons of burned coal.

Estimations were conducted using the MachineLearning Impact calculator presented in Lacoste et al. (2019).

## REPRODUCIBILITY STATEMENT

To allow reproducibility of our results, the code, as well as the hyperparameters used to produce each result presented in this paper, are available at `https://github.com/montrealrobotics/iv_rl`.

The environments and implementation we used (OpenAI Gym, BSuite, MBBL) are all publicly accessible, although a Mujoco license is needed to run some of them.

## ACKNOWLEDGEMENTS

The authors wish to thank the DEEL project CRDPJ 537462-18 funded by the National Science and Engineering Research Council of Canada (NSERC) and the Consortium for Research and Innovation in Aerospace in Québec (CRIAQ). They also acknowledge the support of NSERC as well as the Canadian Institute for Advanced Research (CIFAR).

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

## A    ALGORITHMS IN DETAILS

In this section, we are detailing the different algorithms described in the paper, and explaining the design choices that we made.

### A.1    BOOTSTRAPDQN

BootstrapDQN (Osband et al., 2016), improved with Randomized Prior Function (RPF) (Osband et al., 2018), is an ensemble-based modification of DQN (Mnih et al., 2013). Instead of a $Q$- and a target network, we use a $Q$-ensemble and a target ensemble. It is the base algorithm upon which IV-DQN is built. The detailed steps at training are presented in algorithm 1[7].

**Network architecture**    The $Q$-ensemble and the target ensemble are initialized identically with $N = 5$ neural networks. We used multi-layer perceptrons with 2 ReLU-activated hidden layers of 64 nodes.

**TD update**    Following the implementation of Osband et al. (2016), the networks are paired one to one between the $Q$- and the target networks during the TD update. The system can be considered as several DQNs working in parallel. At training time, the best action $a'$ for each $Q$-network is selected based on the corresponding target network.

---

[7]The presentation of algorithms is optimized for readability. During the implementation, many for-loops are replaced by tensor computation or parallel computing.

**Exploration** During exploration at training time, one network of the ensemble is used to predict the best action for a whole episode. This follows Osband et al. (2016), and is an approximation of Thompson sampling (Thompson, 1933). We added random prior functions (Osband et al., 2018) to tackle the variance underestimation at early training.

At test time, the best action $a'$ is instead selected by a vote from the networks in the ensembles. Ties are broken randomly. Using the mean value across the ensemble was also tested, with slightly lower performances.

**Masked replay buffer** To mask the replay buffer, each step saved in the replay buffer is associated with a $N$-sized boolean vector $m$ which is used as a mask during training. Each element of $m$ indicates if the corresponding network in the $Q$-ensemble should be trained using this sample. These elements of $m$ are independently generated when the sample is saved, using a Bernoulli distribution with probability $p_m$. They are kept fixed during the training. The role of the mask is to ensure variability in the training of the different networks of the $Q$-ensemble and thus maintain a significant variance prediction.

---

**Algorithm 1** BootstrapDQN - Training

---

1: **Input:** RPF scale $\delta_{\mathrm{RPF}}$, minibatch size $K$, mask probability $p_m$, ensemble of $N$ $Q$-networks $\mathcal{Q}$, ensemble of $N$ target-networks $\mathcal{T}$, ensemble of $N$ prior-networks $P$
2: **for** each episode **do**
3:     Pick a $Q$-network from $\mathcal{Q}$ using $i \sim \mathrm{Uniform}\,(1..N)$         ▷ (Osband et al., 2016)
4:     **for** each time step **do**
5:         $a \leftarrow \arg\max_\alpha \left(\mathcal{Q}_i(s, \alpha) + \delta_{\mathrm{RPF}} P_i(s, \alpha)\right)$     ▷ Random prior (Osband et al., 2018)
6:         Collect next-state $s'$ and reward $r$ from the environment by taking action $a$
7:         Sample bootstrap mask $M = m_l \sim \{\mathrm{Bernoulli}(p_m) | l \in 1, ..., N\}$
8:         Add $(s, a, s', r, M)$ in buffer $\mathcal{B}$
9:         $s \leftarrow s'$
10:        Sample minibatch $D$ from replay buffer $\mathcal{B}$
11:        **for** $j$ in $[1 \ldots N]$ **do**     ▷ Each member of the ensemble
12:            **for** each tuple $(s_k, a_k, s'_k, r_k, M_k)$ in minibatch $D$ **do**   ▷ Computing the targets
13:               $\hat{Q}_k \leftarrow \mathcal{Q}_j(a_k, s_k) + \delta_{\mathrm{RPF}} P_j(s_k, a_k))$     ▷ Expected $Q$-value
14:               $a'_k \leftarrow \arg\max_a \left(\mathcal{T}_j(s'_k, a) + \delta_{\mathrm{RPF}} P_j(s'_k, a)\right)$
15:               $T_k \leftarrow r_k + \gamma(\mathcal{T}_j(s'_k, a'_k) + \delta_{\mathrm{RPF}} P_j(s'_k, a'_k))$     ▷ Target
16:            **end for**
17:            Mask minibatch $D$ using $M_j$
18:            Update $\mathcal{Q}_j$'s parameters $\theta_j$ by minimizing $L2$ loss: $(\hat{Q} - T)^2$ over $D$
19:            Soft update of the $\mathcal{T}_j$'s parameters $\bar{\theta}_j \leftarrow (1 - \tau)\bar{\theta}_j + \tau\theta_j$
20:        **end for**
21:     **end for**
22: **end for**

---

### A.2   IV-DQN

IV-DQN is built upon BootstrapDQN. The main differences are the use of var-networks in the ensemble, the computation of the variance in the TD update, and the use of $\mathcal{L}_{\mathrm{IVRL}}$ when training the networks' parameters commonly used to represent epistemic uncertainty in ML. The pseudo-code for IV-DQN is given in algorithm 2, where the differences with BootstrapDQN are highlighted. At test time, action selection for IV-DQN is same as BootstrapDQN.

**Var-networks architecture** Similar to BootstrapDQN, we used ensembles consisting of $N = 5$ networks. The var-networks were also multi-layer perceptrons with 2 ReLU-activated hidden layers of 64 nodes. The only difference is that, instead of a single head for a given $(s, a)$, these networks now provide 2 heads to predict the mean $\mu_\theta$ and the variance $\sigma_\theta^2$ of the $Q$-value. To differentiate these outputs of network $\mathcal{N}$ given a state-action pair $(s, a)$, we write $\mathcal{N}^\mu(s, a)$ and $\mathcal{N}^{\sigma^2}(s, a)$.

**Computing the uncertainty of $\bar{Q}(s', a')$**  As justified in section 3, variance $\sigma^2_{\bar{Q}(s',a')}$ is estimated using var-ensembles. For each $(s'_k, a'_k)$ pair used to train a single $Q$-network $\mathcal{Q}_j$ in ensemble $\mathcal{Q}$, the value of $\bar{Q}(s'_k, a'_k)$ is the mean $\mu_{k,j}$ predicted by the corresponding target-network $\mathcal{T}_j$. However, to evaluate $\sigma^2_{\bar{Q}}$, every target-network $\mathcal{T}_l$ in the target ensemble predicts a mean $\mu_{k,l}$ and a variance $\sigma^2_{k,l}$ for $\bar{Q}(s'_k, a'_k)$. Then, $\sigma^2_{\bar{Q},k}(s'_k, a'_k)$ is computed as the variance of the mixture of Gaussians composed of all the $\mu_{k,l}$ and $\sigma^2_{k,l}$:

$$\sigma^2_{\bar{Q},k} = \frac{1}{N} \sum_{l=1}^{N} \sigma^2_{k,l} + \mu^2_{k,l} - \mu^2_{k,\text{all}} \tag{14}$$

where $\mu^2_{k,\text{all}}$ is the average of all $\mu^2_{k,l}$'s.

**Using $\mathcal{L}_{\text{IVRL}}$**  Given $\sigma^2_{\bar{Q}}$, each network can now learn to predict mean $\mu$ and variance $\sigma^2$ during the TD update using $\mathcal{L}_{\text{IVRL}}$ instead of $L2$. To select the value of $\xi$, we specify a minimal effective batch size $MEBS$. We use equation (2) to determine if, for the values of $\sigma^2_{\bar{Q},k}$ and $\xi = 0$, the effective batch size $EBS$ is lower than $MEBS$. If this is the case, $\xi$ is increased until $EBS \geq MEBS$ using a Nelder Mead optimization method (Nelder & Mead, 1965).

---

**Algorithm 2** IV-DQN - Training. In red, the differences with BootstrapDQN.

---

1: **Input:** RPF scale $\delta_{\text{RPF}}$, minibatch size $K$, mask probability $p_m$, LA weight $\lambda$, minimal effective batch size $MEBS$, initial state $s$, var-ensemble of $N$ $Q$-var-networks $\mathcal{Q}$, var-ensemble of $N$ target-var-networks $\mathcal{Q}$, var-ensemble of $N$ prior-var-networks $P$
2: **for** each episode **do**
3:      Pick a $Q$-network from $\mathcal{Q}$ using $i \sim \text{Uniform}(1..N)$      ▷ (Osband et al., 2016)
4:      **for** each time step **do**
5:          $a \leftarrow \arg\max_\alpha (\mathcal{Q}_i(s, \alpha) + \delta_{\text{RPF}} P_i(s, \alpha))$    ▷ Random prior (Osband et al., 2018)
6:          Collect next-state $s'$ and reward $r$ from the environment by taking action $a$
7:          Sample bootstrap mask $M = m_l \sim \{\text{Bernoulli}(p_m) | l \in 1, ..., N\}$
8:          Add $(s, a, s', r, M)$ in buffer $\mathcal{B}$
9:          $s \leftarrow s'$
10:         Sample minibatch $D$ from replay buffer $\mathcal{B}$
11:         **for** $j$ in $[1 \ldots N]$ **do**
12:             **for** each tuple $(s_k, a_k, s'_k, r_k, M_k)$ in mini-batch $D$ **do**
13:                $\mu_{\hat{Q},k} \leftarrow \mathcal{Q}_j^\mu(a_k, s_k) + \delta_{\text{RPF}} P_j(s_k, a_k))$      ▷ Expected $Q$-value
14:                $a'_k \leftarrow \arg\max_a (\mathcal{T}_j(s'_k, a) + \delta_{\text{RPF}} P_j(s'_k, a))$
15:                $T_k \leftarrow r_k + \gamma(\mathcal{T}_j(s'_k, a'_k) + \delta_{\text{RPF}} P_j(s'_k, a'_k))$      ▷ Target
16:                **for** $l$ in $[1 \ldots N]$ **do**      ▷ $\mu$ and $\sigma^2$ for each network of the ensemble
17:                    $\mu_{k,l} \leftarrow \mathcal{T}_l^\mu(s'_k, a'_k) + \delta_{\text{RPF}} P_l^\mu(s'_k, a'_k)$
18:                    $\sigma^2_{k,l} \leftarrow \mathcal{T}_l^{\sigma^2}(s'_k, a'_k)$
19:                **end for**
20:                $\mu_{k,\text{all}} \leftarrow 1/N \sum_{l=1}^{N} \mu_{k,l}$      ▷ Average of the mean predictions
21:                $\sigma^2_{\bar{Q},k} \leftarrow 1/N \sum_{l=1}^{N} \sigma^2_{k,l} + \mu^2_{k,l} - \mu^2_{k,\text{all}}$      ▷ Variance of mixture of Gaussians
22:             **end for**
23:             Estimate $\xi$ based on all $\sigma^2_{\bar{Q},k}$ so that $EBS \geq MEBS$ using Eq. (1)
24:             Mask minibatch using $M_j$
25:             Train $\mathcal{Q}_j$'s parameters $\theta_j$ with $\mathcal{L}_{\text{IVRL}}$ (10) using $\mu_{\hat{Q}}$, $T$, $\xi$ and $\sigma^2_{\bar{Q}}$ on $D$
26:             Soft update of $\mathcal{T}_j$'s parameters $\bar{\theta}_j \leftarrow (1 - \tau)\bar{\theta}_j + \tau\theta_j$
27:         **end for**
28:      **end for**
29: **end for**

---

### A.3  ABLATIONS FOR IV-DQN

In the ablation study presented in section 4.1, 4 new algorithms are presented. We detail here their differences with IV-DQN and algorithm 2.

**BIV + BootstrapDQN**   In BIV + BootstrapDQN, the networks used in the ensembles are single-head networks, similar to BootstrapDQN. Thus, these are sampled ensembles. $\sigma_{\bar{Q},k}^2$ is thus computed as the sampled variance of the networks' predictions. Additionally, instead of $\mathcal{L}_{\text{IVRL}}$, we simply use $\mathcal{L}_{\text{BIV}}$.

Compared to algorithm 2, lines 1 to 17 are the same. Line 18 is removed. Line 21 is changed by: $\sigma_{\bar{Q},k}^2 \leftarrow 1/N \sum_{l=1}^{N}(\mu_{\bar{Q},l} - \mu_{\bar{Q},\text{all}})^2$. Finally, in line 25, $\mathcal{L}_{\text{IVRL}}$ is changed to $\mathcal{L}_{\text{BIV}}$.

**L2 + VarEnsembleDQN**   In L2 + VarEnsembleDQN, we do not apply BIV weights. The difference with IV-DQN is that we simply change $\mathcal{L}_{\text{BIV}}$ for the $L2$ loss in $\mathcal{L}_{\text{IVRL}}$. The networks are still var-networks trained with the $\mathcal{L}_{\text{LA}}$ part of the $\mathcal{L}_{\text{IVRL}}$ loss, and the rest of the algorithm is the same as algorithm 2. In practice, the computation of $\sigma_{\bar{Q},k}^2$ and $\xi$ are now useless, and lines 16 to 21 as well as line 23 can be removed.

**BIV + VarNetworkDQN**   In BIV + VarNetworkDQN, we use a single var-network instead of a var-ensemble. Concretely, $N = 1$. This means that at line 21, $\sigma_{\bar{Q},k}^2$ is effectively the variance output by the single target network.

Additionally, masks and RPF do not make sense here, so they are not applied. As the BootstrapDQN exploration cannot apply either, an $\epsilon$-greedy strategy is used instead.

**L2 + VarNetworkDQN**   L2 + VarNetworkDQN is BIV + VarNetworkDQN, without BIV weights. $\mathcal{L}_{\text{BIV}}$ is replaced by the $L2$ loss in $\mathcal{L}_{\text{IVRL}}$.

## A.4   IV-SAC

IV-SAC is built upon EnsembleSAC, which is inspired by SUNRISE (Lee et al., 2021). We describe here the main elements of IV-SAC, and present the detailed process in algorithm 3.

**Network architecture**   The variance networks used in IV-SAC are multilayer perceptrons with flattened inputs and 2 ReLU-activated hidden layers of 256 nodes. Our implementation is based on rlkit[8].

**Critic**   SAC is a complex DRL algorithm with several details which are orthogonal to IV-RL, for example, the accounting of policy entropy in the definition of the $Q$-value. On the critic side, the implementation of IV-RL is mostly similar to IV-DQN's TD update.

**Actor**   Following the example of Lee et al. (2021), we use an ensemble for the actor-network when using ensembles for the critic. Each network in the actor is linked to a pair of networks in the critic. When training, the $Q$-value and its variance for each state-action are computed similarly as when training the critic. As justified in section 3.2, we also apply the BIV loss function when training the actor. However, instead of the $L2$ error shown in equation (1), we use the SAC loss function for the actor. For a minibatch $D$ of size $K$, we have:

$$\mathcal{L}_{\text{actor}}(D, \phi) = \left(\sum_{k=0}^{K} \frac{1}{\sigma_{\hat{Q}}^2(s_k, a_k) + \xi}\right)^{-1} \sum_{k=0}^{K} \frac{1}{\sigma_{\hat{Q}}^2(s_k, a_k) + \xi}(\alpha \ln(\pi_\phi(a_k|s_k)) - \mu_{\hat{Q}}(s_k, a_k)) \tag{15}$$

**Action selection**   The use of an ensemble for the d allows for different methods to select the next action given a state. In this respect, we follow again the example of Lee et al. (2021).

During the training of the Q-network, we pair each Q-network with its corresponding actor. To encourage exploration, we additionally take advantage of the ensemble to implement UCB. Each policy network outputs an action. The mean $Q$-value $\mu_Q(s, a)$ and variance $\sigma_Q^2(s, a)$ for each action

---

[8]https://github.com/rail-berkeley/rlkit

$a$ are computed by sampling over all networks of the critic. The chosen action has the highest UCB score: $a^* = \arg\max_a[\mu_Q(s,a) + \Lambda_{\text{UCB}}\sigma_Q(s,a)]$, where $\Lambda_{\text{UCB}}$ is a hyperparameter.

To estimate the standard deviation for UCB, we use sampled ensembles: this is because, for UCB, the variance estimation must be calibrated, in order to be correctly scaled with the $Q$-values. Var-ensembles are therefore disqualified.

During testing, the agent takes the average action, as sampled from the output of the mean prediction by the policy networks.

---

**Algorithm 3** IV-SAC - Training

---

1: **Input:** LA weight $\lambda$, UCB $\Lambda_{\text{UCB}}$, minimal effective batch size $MEBS$, policy networks $\pi$, ensemble of $N$ $Q$-var-networks $\mathcal{Q}$, ensemble of $N$ target-var-networks $\mathcal{T}$
2: **for** each episode **do**
3:      **for** each time step **do**
4:          Collect the action samples: $\mathcal{A} \leftarrow \{a_i \sim \pi_i(a|s)|i \in \{1,...,N\}\}$
5:          $a \leftarrow \arg\max_{a_i \in \mathcal{A}}[\mu(\mathcal{Q}(s,a_i)) + \Lambda_{\text{UCB}}\sigma(\mathcal{Q}(s,a_i))]$        $\triangleright$ UCB exploration
6:          Collect next-state $s'$ and reward $r$ from the environment by taking action $a$
7:          Sample bootstrap mask $M \leftarrow \{m_l \sim \text{Bernoulli}(p_m)|l \in \{1,...,N\}\}$
8:          Store tuple $(s, a, s', r, M)$ in buffer $\mathcal{B}$
9:          $s \leftarrow s'$
10:      **end for**
11:      **for** each optimization step **do**
12:          Sample minibatch $D$ from the replay buffer $\mathcal{B}$
13:          **for** $j$ in $[1 \ldots N]$ **do**        $\triangleright$ Each member of the ensemble
14:              **for** each tuple $(s_k, a_k, s'_k, r_k, M_k)$ in minibatch $D$ **do**    $\triangleright$ Computing the targets
15:                  Sample next-action: $a'_{k,j} \sim \pi_j(a'|s'_k)$
16:                  Compute Expected Q-value: $\mu_{\hat{Q},k} \leftarrow \mathcal{Q}_j^\mu(a_k, s_k)$
17:                  Compute Target: $T_k \leftarrow r_k + \gamma\mathcal{T}_j^\mu(s'_k, a'_{k,j})$
18:                  $\sigma^2_{\hat{Q},k} \leftarrow$ Predictive-Variance$(\mathcal{Q}, s_k, a_{k,j})$          $\triangleright$ Algorithm 4
19:                  $\sigma^2_{\bar{Q},k} \leftarrow$ Predictive-Variance$(\mathcal{T}, s'_k, a'_{k,j})$          $\triangleright$ Algorithm 4
20:              **end for**
21:              Mask minibatch $D$ using $M_j$
22:              Estimate $\xi_a$ so that $EBS(\sigma^2_{\hat{Q}}) \geq MEBS$ using Eq. (2)
23:              Update $\pi_j$'s parameters $\phi_j$ with $\mathcal{L}_{\text{actor}}$ (15) using $\mu_{\hat{Q}}, \pi_j, \xi_a$ and $\sigma^2_{\hat{Q}}$
24:              Estimate $\xi_c$ so that $EBS(\sigma^2_{\bar{Q}}) \geq MEBS$ using Eq. (2)
25:              Update $\mathcal{Q}_j$'s parameters $\theta_j$ with $\mathcal{L}_{\text{IVRL}}$ (10) using $\mu_{\hat{Q}}, T, \xi_c$ and $\sigma^2_{\hat{Q}}$ on $D$
26:              Soft update of $\mathcal{T}_j$'s parameters $\bar{\theta}_j \leftarrow (1-\tau)\bar{\theta}_j + \tau\theta_j$
27:          **end for**
28:      **end for**
29: **end for**

---

---

**Algorithm 4** Variance Estimation for IV-SAC

---

1: **procedure** PREDICTIVE-VARIANCE$(\mathcal{N}, s, a)$
2:      **Input:** Var-networks ensemble: $\mathcal{N}$, State: $s$, Action: $a$
3:      **for** $l$ in $\{1...N\}$ **do**
4:          $\mu_l \leftarrow \mathcal{N}_l^\mu(s,a)$
5:          $\sigma_l^2 \leftarrow \mathcal{N}_l^{\sigma^2}(s,a)$
6:      **end for**
7:      $\mu_\mathcal{N} \leftarrow 1/N \sum_{l=1}^N \mu_l$          $\triangleright$ Average of the mean predictions
8:      $\sigma_\mathcal{N}^2 \leftarrow 1/N \sum_{l=1}^N \sigma_l^2 + \mu_l^2 - \mu_\mathcal{N}^2$          $\triangleright$ Variance of mixture of Gaussians
9:      **return** $\sigma_\mathcal{N}^2$
10: **end procedure**

---

## A.5 ABLATIONS FOR IV-SAC

We also do ablation studies with our IV-SAC model (presented in section 4.2) where we show 6 additional variations. We detail here their differences with IV-SAC with respect to algorithm 3.

**EnsembleSAC**   EnsembleSAC is the base on which IV-SAC is built. It uses regular networks as critics instead of variance networks in the ensemble, and it does not consider uncertainty. Compared to algorithm 4, lines 1 to 17, and 20, 21, 26 to 29 are identical. Lines 18, 19, 22 and 24 are removed. In line 23 and 25, the usual losses of SAC are used instead of $\mathcal{L}_{\text{actor}}$ and $\mathcal{L}_{\text{BIV}}$.

**BIV+EnsembleSAC**   Similar to EnsembleSAC, this version uses regular networks as critic instead of variance networks in the ensemble. However, it is uncertainty aware: it estimates the variance of the prediction using sampled ensembles. It is thus similar to SunriseSAC (Lee et al., 2021), but with BIV weights instead of SUNRISE weights. $\sigma^2_{\hat{Q},k}$ and $\sigma^2_{\bar{Q},k}$ being computed as the sampled variance of the networks' predictions, we change these lines compared to algorithm 3:

- line 18 to: $\sigma^2_{\hat{Q},k} \leftarrow 1/N \sum_{l=1}^{N} (\mu_{\hat{Q},l} - 1/N \sum_{m=1}^{N} \mu_{\hat{Q},m})^2$

- line 19 to: $\sigma^2_{\bar{Q},k} \leftarrow 1/N \sum_{l=1}^{N} (\mu_{\bar{Q},l} - 1/N \sum_{m=1}^{N} \mu_{\bar{Q},m})^2$

Additionally, instead of $\mathcal{L}_{\text{IVRL}}$, we use $\mathcal{L}_{\text{BIV}}$ for the critic at line 25.

**L2 + VarEnsembleSAC**   Here, we do not apply BIV weights to both actor and critic losses. The critic networks are still var-networks trained with the $\mathcal{L}_{\text{LA}}$ part of the $\mathcal{L}_{\text{IVRL}}$ loss, and the rest of the algorithm is the same as algorithm 3. In practice, the computation of $\sigma^2_{\hat{Q},k}$ and $\xi$ are now useless, and lines 18, 19, 22 and 24 can be removed.

**BIV + VarNetworkSAC**   We use a single var-network instead of a var-ensemble as the critic. Concretely, $N = 1$. This means that at lines 18 and 19, $\sigma^2_{\hat{Q},k}$ and $\sigma^2_{\bar{Q},k}$ are effectively the variance output by the single-target network.

**L2 + VarNetworksAC**   VarNetwork is BIV + VarNetwork, without BIV weights in both actor and critic losses. $\mathcal{L}_{\text{BIV}}$ is replaced by the $L2$ loss in $\mathcal{L}_{\text{IVRL}}$.

**UWAC + VarEnsemble**   Here we change the weighing scheme in IV-SAC to the one proposed in Wu et al. (2021), keeping everything else the same.

**Sunrise + VarEnsemble**   For this version we modify the weighing scheme in IV-SAC to the one proposed in Lee et al. (2021), keeping everything else the same.

## B   HYPERPARAMETER TUNING

Deep reinforcement learning algorithms are known to be particularly difficult to evaluate due to the high stochasticity of the learning process and the dependence on hyperparameters. To thoroughly compare different algorithms, it is important to carefully tune the hyperparameters, based on several environments and network initialization seeds (Henderson et al., 2018).

For every result presented in this paper, the hyperparameters for each algorithm were tuned using grid search. Each combination of hyperparameters was run five times, with different seeds on both initialization and environment. The best 3 or 4 configurations were then selected to be run 25 times - this time, the combinations of 5 environment and 5 initialization seeds. The configuration selected to be shown in the paper is the best of these configurations based on the 25 runs.

Table 3 describes the type and sets of parameters that were optimized for the relevant algorithms. Note that the learning rate $lr$, discount rate $\gamma$ and target new update rate $\tau$ are also relevant for SAC. Because there was too many hyperparameters to tune, we kept for all SAC experiments the

Table 3: List and set of hyperparameters that were tuned when testing the algorithm.

| Hyperparameter | Relevant algorithms | Set |
|---|---|---|
| Effective minibatch size | All | $\{32, 64, 128, 256\}$ for DQN+
$\{256, 512, 1024\}$ for SAC+ |
| Learning rate $lr$ | DQN+ | $\{0.0005, 0.001, 0.005, 0.01\}$ |
| Discount rate $\gamma$ | DQN+ | $\{0.98, 0.99, 0.995\}$ |
| Target net update rate $\tau$ | DQN+ | $\{0.001, 0.005, 0.01, 0.05\}$ |
| $e$-decay | DQN+ with
$e$-greedy exploration | $\{0.97, 0.98, 0.99, 0.991, 0.995\}$ |
| Mask probability | All with masked
ensembles | $\{0.5, 0.7, 0.8, 0.9, 1\}$ |
| RPF prior scale | All with RPF | $\{0.1, 1, 10\}$ |
| Loss attenuation weight $\lambda$ | All with loss attenuation | $\{0.01, 0.1, 0.5, 1, 2, 5\}$ |
| Minimal EBS ratio | All with minimal EBS | $\{16/32, 24/32, 28/32, 30/32, 31/32\}$ for DQN+
$\{0.5, 0.9, 0.95, 0.99\}$ for SAC+ |
| Minimal variance $\epsilon$ | All with constant $\epsilon$ | $\{0.5, 1, 5, 100, 1000, 10000\}$ |
| UCB weight $\lambda_{\text{UCB}}$ | All SAC+ with UCB
(0 means Bootstrap) | $\{0, 1, 10\}$ as suggested in Lee et al. (2021) |
| SUNRISE temperature | All with SUNRISE | $\{10, 20, 50\}$ as suggested in Lee et al. (2021) |
| UWAC $\beta$ | All with UWAC | $\{0.8, 1.6, 2.5\}$ as suggested in Wu et al. (2021) |

default values used in *rlkit*[9] implementation : $actorlr = 0.0003$, $criticlr = 0.0003$, $\gamma = 0.99$ and $\tau = 0.005$.

The hyperparameters used for each curve can be found in the configuration file of the code submitted as additional material.

## C   $\lambda$ IN $\mathcal{L}_{\text{IVRL}}$

As expressed in equation (10), the $\mathcal{L}_{\text{IVRL}}$ loss function we use to train the $Q$-value estimation is a combination of two components, $\mathcal{L}_{\text{BIV}}$ and $\mathcal{L}_{\text{LA}}$, balanced by hyperparameter $\lambda$.

We use Adam (Kingma & Ba, 2015) optimizer: any constant multiplied to the loss function is absorbed and does not impact the gradient. Therefore, $\lambda$ effectively controls the ratio between $\mathcal{L}_{\text{BIV}}$ and $\mathcal{L}_{\text{LA}}$.

The sensitivity to the value of $\lambda$ depends on the scale of the return, the stochasticity of the environment, and the learning dynamics. As can be seen in figure 6, changing the values of $\lambda$ between 1 and 10 lead to different results in LunarLander, HalfCheetah and Ant, while in SlimHumanoid, Walker and Hopper, the performance stays very similar.

In figure 7, we analyze how the performance of IV-RL methods change when $\lambda$ is set to extreme values. Very high values of $\lambda$ lead to sub-optimal results, first because $\mathcal{L}_{\text{BIV}}$ is practically ignored, but also because using variance ensembles alone may lead to a slower or less stable learning curve. Very low values of $\lambda$, on the other hand, also bring the performance down, first as $\mathcal{L}_{\text{LA}}$ is ignored, but also as the variance prediction is not trained: the uncertainty estimates provided to $\mathcal{L}_{\text{BIV}}$ are less reliable.

From figure 7, we can see that the effect of $\lambda$ depends on the environment. For LunarLander, smaller $\lambda$ leads to slower learning and sub-optimal performance in comparison to $\lambda = 10$. On the other hand, very high $\lambda$ leads to poor performance and never reach the goal score of 200. For SlimHumanoid, both extremes show inferior performance to when $\lambda = 10$, but, although there are differences in the dynamics, they reach similar a score after 200k steps. In Walker, high $\lambda$ is also slower, however, the impact of low $\lambda$ is less important. This can be attributed to the fact that, for Walker, the BIV component contributes more to performance than the LA component of IV-RL loss, as seen in figure 5.

---

[9]https://github.com/rail-berkeley/rlkit

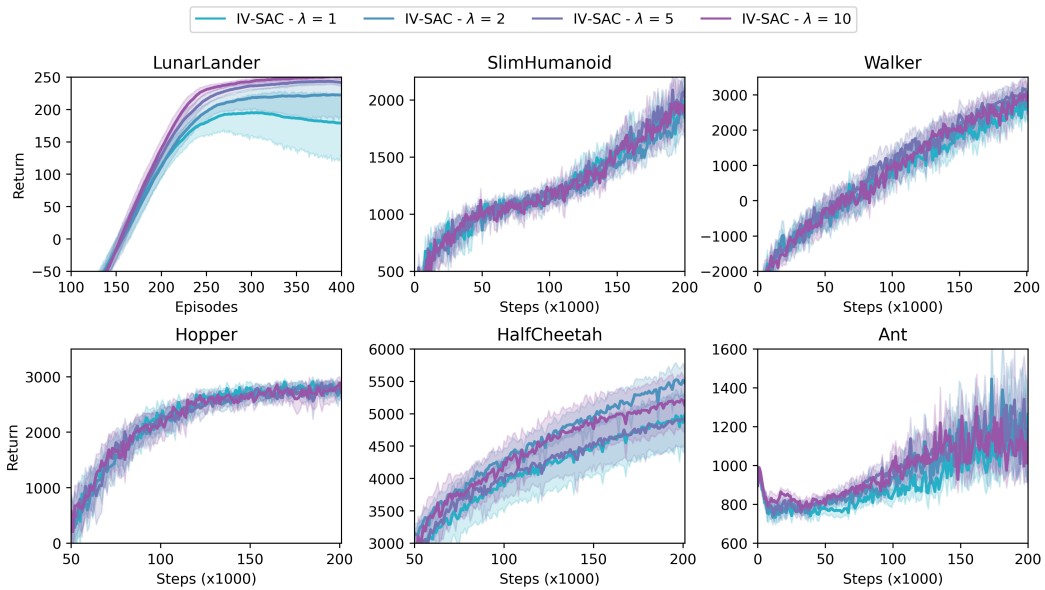

Figure 6: IV-RL with $\lambda$ values around the range of 1 to 10.

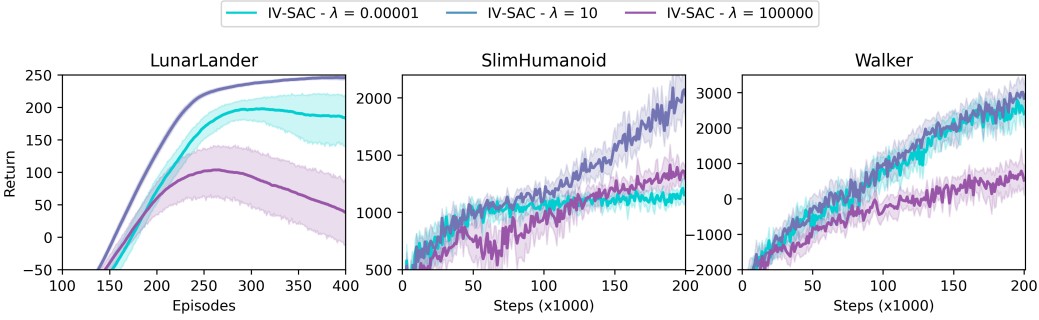

Figure 7: IV-RL with extreme values of $\lambda$

# D  COMPUTATION TIME

As IV-RL uses ensembles, the computation time necessary for the training of the agents is significantly increased compared to single-network algorithms such as SAC or DQN. This can be seen in tables 4 and 5.

The computation of $\xi$ for $MEBS$ as well as additional forward passes to evaluate the variance add some computational load compared to other ensemble-based methods such as BootstrapDQN and SunriseDQN.

While this can limit the applicability of IV-RL, we consider the cases where computation time is secondary to sample efficiency. This is the case when samples are the more expensive element, such as in the real world.

Table 4: Average training time for 1000 steps on walker2d environment.

|  | SAC | EnsembleSAC | SunriseSAC | IV-SAC (ours) |
|---|---|---|---|---|
| Runtime/1000 steps (s) | 8.34 | 62.16 | 62.1 | 74.56 |

Table 5: Average training time per episode on LunarLander-v2 environment

|  | DQN | BootstrapDQN | SunriseDQN | IV-DQN (ours) |
| --- | --- | --- | --- | --- |
| Runtime/ episode (s) | 0.51 | 1.5 | 1.53 | 1.87 |

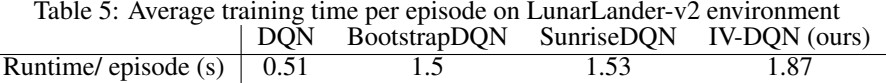

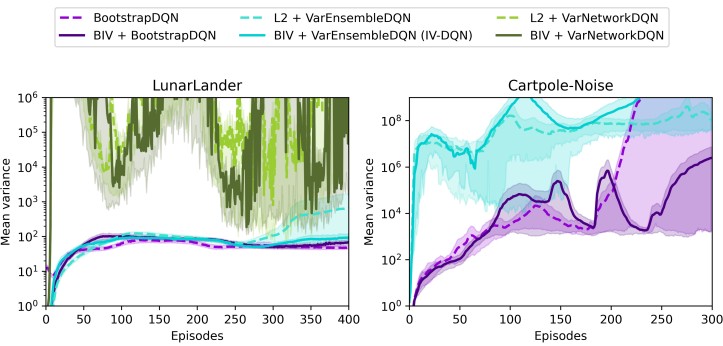

Figure 8: Mean target variance predictions for discrete environments

# E VARIANCE ESTIMATION

The difference of performance between IV-RL with sampled ensembles and variance ensembles is sometimes significant. Looking at the variance prediction can allow us to better understand this result.

## E.1 IV-DQN

In figure 8, we show the mean of the sample variance estimations per mini-batch, depending on the variance estimation method.

Different profiles can be seen in LunarLander and Cartpole-Noise. In the first environment, the scale of the variance predicted by sampled ensembles with BootstrapDQN and variance ensembles does not differ significantly. In Cartpole-Noise however, the variance ensembles quickly capture much more uncertainty. This explains the difference in dynamics that can be seen in figure 3.

Although we do not have results for single variance networks in Cartpole-Noise, we can see in LunarLander why single variance networks are less reliable than variance ensembles. The mean variance they predict is very high, but, as can be seen in figure 9, the median is a lot lower. The mean is thus driven by extremely high variance predictions. Instead, sampled and variance ensembles show similar profiles for mean and median variance. The low median variance confirms that single variance networks fail to capture epistemic uncertainty, but also that they have very high volatility.

## E.2 IV-SAC

In the continuous control environments, the story is different, as can be seen in figures 10 and 11.

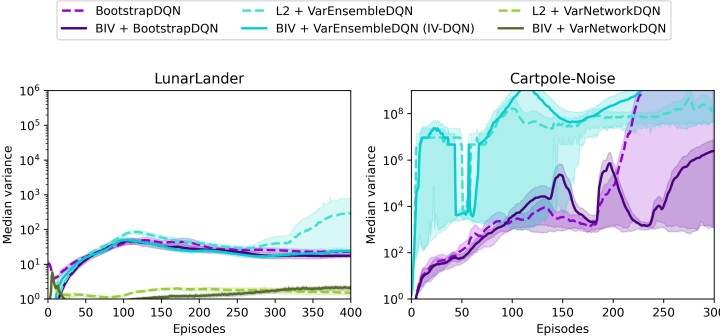

Figure 9: Median target variance predictions on discrete environments

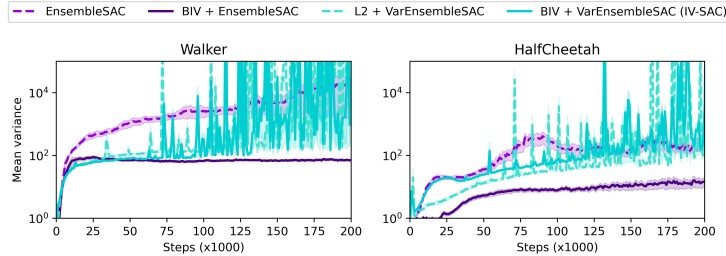

Figure 10: Mean target variance predictions on continuous environments

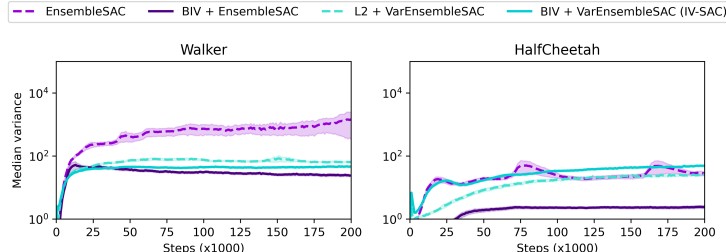

Figure 11: Median target variance predictions on continuous environments

The first thing to notice is the volatility of the mean of the variance produced by variance ensembles. Although the scale is not completely unreasonable and is still close to the median, (on the opposite of variance networks in DQN), it still suffers from some calibration issues. Second, we see that the epistemic variance in EnsembleSAC is higher than the one in BIV + EnsembleSAC. One hypothesis to explain this is that using IV reduces the impact of very noisy supervision which may confuse the networks and increase the variance in the ensemble. Variance ensembles do not seem to suffer from this.

## F    WEIGHTING SCHEMES

The study on the variance allows a better understanding of the advantage of the BIV weights (Mai et al., 2021) compared to the UWAC weights (Wu et al., 2021) and SUNRISE weights (Lee et al., 2021).

In these works, the weight $w_k$ of a sample $k$ in a minibatch of size $K$ is given by:

$$w_{\text{BIV},k} = \left( \sum_{j=1}^{K} \frac{1}{\sigma_j^2 + \xi} \right)^{-1} \frac{1}{\sigma_k^2 + \xi} \tag{16}$$

$$w_{\text{UWAC},k} = 1/K \min \left( \frac{\beta}{\sigma_k^2}, 1.5 \right) \tag{17}$$

$$w_{\text{SUNRISE,k}} = 1/K \text{Sigmoid}(-\sigma_k * T) + 0.5 \tag{18}$$

Figure 12 shows the result of applying such weights to variance ensembles. Note that the first two plots are repeated from figure 5 in section 4.2.

It is clear that the SUNRISE weights, which are based on a heuristic, are not optimal compared to the more theoretically grounded UWAC and BIV weights.

However, we can note that UWAC weights perform much better than SUNRISE, but are still somewhat inferior to BIV weights. Our understanding is that this is due to the changes in the scale of the variance estimation along the learning process, as seen in figures 8 and 10.

In UWAC, this first leads to continuous changes in the effective learning rate. For example, multiplying the variances by 2 divides the weights accordingly by 2. While the multiplication of the loss by a constant is captured by optimizers such as Adam (Kingma & Ba, 2015) in the long term, it can

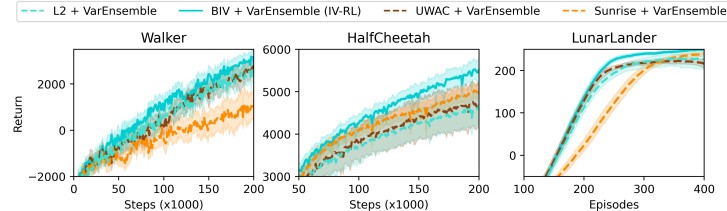

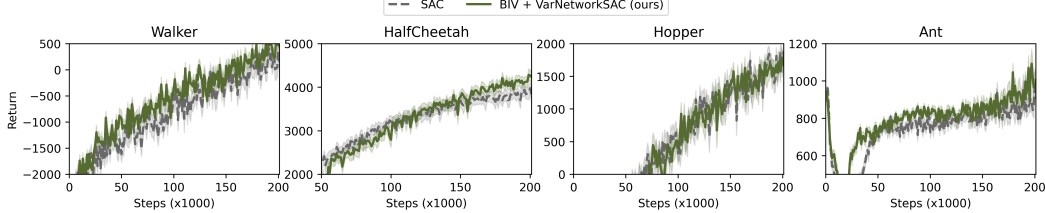

Figure 12: Comparing BIV, UWAC and SUNRISE weights with var-ensemble

Figure 13: Single variance networks combined with BIV lead to a very slight improvement.

still cause short term disturbances in the learning process. In comparison, the normalization of the BIV weights ensures the stability of the loss function.

UWAC weights are also hard-clipped by the value 1.5, after being scaled by a constant hyperparameter $\beta$. When the scale of the variances changes, this can disturb the control over the discriminatory power of the weights. For example, if all variances are smaller than $\beta$, all weights are clipped and become identical at 1.5, and the weighting scheme does not discriminate between certain and uncertain samples. On the other hand, if most variances are very high compared to $\beta$, the weights will be very low. A single low-variance sample may thus completely outweigh all other samples, which will get ignored in the gradient update. The discrimination is then too strong. BIV instead uses $\xi$, which is dynamically computed to maintain a minimal effective batch size $MEBS$ as explained in section 2.1. The relative weights of the samples are thus preserved if the variances are multiplied by a constant. As such, the BIV weights maintain a stable discriminatory power in the face of changing variance scales along the learning process, leading to better results.

Note that the curves here present the combination of UWAC and SUNRISE with var-ensembles, which is different from the original works proposed by Wu et al. (2021) (using MC-Dropout) and Lee et al. (2021) (using sampled ensembles).

## G  VARIANCE NETWORKS

What if, instead of using predictive ensembles, we simply replaced the $Q$-network by a variance network and trained it using the IV-RL loss?

Figure 13 shows the performances on 4 continuous environments, compared to SAC. We can see a slight improvement, but it is not statistically significant. Based on the insights given in section E, we make the hypothesis that it is due to the volatility of the variance estimation with a single variance network.

## H  IV-RL AND OPTIMISM IN THE FACE OF UNCERTAINTY

A question that arose during this research is how does IV-RL interact with the principle of Optimism in the Face of Uncertainty (OFU), which is often used to drive exploration in reinforcement learning, for example with Upper Confidence Bounds (UCB).

The principle behind IV-RL is to down-weight highly uncertain samples during the TD-update; the idea of OFU is to encourage the agent to choose actions with high uncertainty during exploration. Do these two approach go against each other?

First, we would like to distinguish the uncertainty of the *value* estimation at state-action pair $Q(s, a)$ used by OFU, and the uncertainty of the value estimation of the *target* $T(s, a)$ used by IV-RL. They are separate: we tell the agent to go where the $Q(s, a)$ is uncertain, and then to sample a target $T(s, a)$ by projecting over the next time step with $Q(s', a')$. $T(s, a)$ has its own uncertainty, which is not that of $Q(s, a)$. If the agent believes $T(s, a)$ is highly uncertain, it should not give it too much weight. $T(s, a)$ may also have low uncertainty: the agent will then give it more importance.

However, two factors come and blur this distinction. First, in the context of reinforcement learning where the experience is built with trajectories, the information the agent has received at $(s', a')$ is correlated to the information it received at $(s, a)$. Second, the use of function approximation may cause the uncertainty of $Q(s, a)$ and the uncertainty of $Q(s', a')$ to be similar, especially if $(s, a)$ is close to $(s', a')$ on the state-action space. These correlations may induce interactions between IV-RL and OFU, which may not be completely decoupled.

Another element needs to be considered: in both IV-DQN and IV-SAC, exploration is made following the $Q$-networks, whereas the value of $Q(s', a')$ to compute the target is estimated by target networks, which are delayed compared to $Q$-networks. This reduces the correlations between the uncertainties of $Q(s, a)$ and $T(s, a)$.

As an empirical element to this answer, we show in the paper that OFU and IV-RL work together: the IV-SAC experiments include a UCB bonus for exploration. It is to be noted that, if these two strategies can conflict, IV-RL proposes the use of hyperparameter $MEBS$ which allows us to control the intensity of its discrimination. By increasing the $MEBS$, one can give more importance to high-variance samples, and find an optimal balance.

Nevertheless, we believe this question is important for all strategies based on weighting samples according to variance in DRL and could be an interesting line of future investigation.

