# OpenReview forum: "Sample Efficient Deep Reinforcement Learning via Uncertainty Estimation"
_ICLR.cc/2022/Conference — ICLR 2022 Spotlight_

### Official Review · Reviewer_Ncaq · 2021-10-30

**Correctness:** 3
**Technical Novelty And Significance:** 4
**Empirical Novelty And Significance:** 3
**Recommendation:** 8
**Confidence:** 4

**Main Review:**

Overall the paper is very well written with clear motivations for the modeling choices used to develop the proposed IV-RL algorithms. The contributions listed at the end of Section 1 are each interesting and valid improvements to the current literature seeking to develop robust Deep RL paradigms. The paper covers a lot of supporting material used to ground the proposed approach and appropriately covers sufficient technical details to appropriately frame the contributions put forward. In particular, I enjoyed the connections drawn to frame TD learning as regression on heteroskedastic noisy labels. This insight drove home the necessity of using BIV to account for the uncertainty induced by approximating the Q function.

*This however leads to the first major weakness in the paper.* It is unclear, as presented at the end of Section 3.1.2, what exact quantities or network structures (ensembles and etc.) are used in BIV and LA exactly. As good as the framing and motivation are, there is a major lack of specificity or explicit formulation of the objectives used to learn the parameters of the policies. The paper would be significantly improved if more careful and direct development of the IV-RL loss was presented. In particular, what forms the estimate for $\sigma_{\bar{Q}}^2$? Is it an ensemble of networks? How does that connect to the various terms in Equation 1? What is the Loss function in the BIV Loss? Is it the standard TD-error? In the end, it would be very helpful if the separate components which were introduced in context of the grounding literature were extended to more explicitly communicate how they fit into the DRL pipeline. Clarifying this component is one of two major weaknesses that, if sufficiently addressed, will motivate me to raise my score.

A minor weakness connected to the lack of clarity just described is that the description of the various baselines are quite vague. It's not clear what is meant by a "non-IV version of the improvements of IV-RL" nor what the ablations in Figure 3 are really showing. The abbreviations/acronyms included in Figures 3 and 5 are not that informative. Again, being more specific here will make the following experimental analyses much easier to understand.

*The second major weakness* that I want to point out is that there it is not abundantly clear what the major emphasis of IV-RL is. At the outset of the paper the Introduction positions the work as different from the DRL literature harnessing uncertainty to investigate exploration/exploitation tradeoffs, emphasizing that this work focuses instead on the quality of supervision while maximizing the expected reward. However, much of the discussion in Section 4 (Experimental Results) uses concepts of exploration or lack thereof as a justifying reason behind some of the presented performance. This inconsistency dilutes the major contributions and significance of the paper as outlined in Section 1. This lack of precision affects the major takeaways and is evident in the Conclusion as there are very few concrete points that motivate further use of IV-RL. I was hoping to read what the authors felt their proposed IV-RL approach was particularly well suited for, what types of problems they feel are easier to solve because of the advances put forward in the paper. In addition to more concrete conclusions, it would be really nice if the authors were able to express limitations to the proposed framework as currently implemented. Are there particular inefficiencies and computational overhead that need to be considered before implementing IV-RL? What other weaknesses are there?

Beyond these specific points I do want to commend the authors for their extensive and rigorous empirical analysis when comparing to baseline and ablated algorithms. The presented results are convincing and there is some decent discussion (it could be sufficiently tightened a little bit as discussed above) to elaborate the findings. Overall, it appears that the proposed IV-RL algorithm works quite well and improves over prior approaches.

**Summary Of The Paper:**

This paper incorporates inverse variance weighting and probabilistic ensembles to account for sources of uncertainty present in the standard TD and Actor Critic learning paradigms. The authors present a systematic analysis of these sources of uncertainty and show how TD learning contributes heteroskedastic behavior, motivating the use of IV weighting of the samples used in agent training. Utlimately, a combination of complementary approaches is used to account for the variance in returns encountered in both the environment and approximation of the Q-Network. The proposed IV-RL algorithms are evaluated in both simple continuous control as well as some MuJoCo tasks, with favorable performance improvements over related prior approaches.

**Summary Of The Review:**

Fantastic framing and decomposition of the sources of uncertainty present in learning an RL policy. The paper introduces inverse variance weighting to RL and does so in an interesting and well motivated manner. Despite some weaknesses in firmly anchoring the foundational pieces of the proposed IV-RL approach, the empirical results are shown to improve over comparative algorithms across a variety of domains. As expressed in my main review, greater clarity about the contributions of each component of the IV-RL loss, framed in a more specific terms, as well as more consistent focus on the proposed contributions throughout the paper will help me raise my assessment from borderline acceptance to full acceptance.

---

> ### Author Response · Authors · 2021-11-18
> **Response to Reviewer Ncaq**
>
> We thank the reviewer for his/her detailed review, and very helpful feedback! With your comments, we were able to improve the paper, especially with respect to clarity of the IV-RL loss and consistency of the contributions.
>
> **Clarity of the IV-RL loss, algorithms, and baselines**
>
> We have worked on several parts to improve the clarity of these elements:
> - We have modified equation 10 to make it more explicit, and clarify the origin of the elements that it contains.
> - Additionally, we provide in Appendix A a detailed pseudo-code for IV-SAC and IV-DQN and also provide implementation details for the baselines.
> - The description of the ablations shown in figures 3 and 5 has been clarified, and
> - We also provide detailed descriptions of the ablations in Appendix A.
>
> We hope these additional details improve the clarity of the paper.
>
> **Emphasis on exploration in the results**
>
> Applying IV-RL to MountainCar did not result in any improvement, because it is an exploration-based environment where the exploration strategy is the main factor in performance.  While we wanted to underline this in the text, we agree that it was causing confusion with respect to the contributions of the paper. We are grateful that you raised this important point.
>
> We have therefore *modified the part of the discussion in section 4.1* that was focusing on exploration strategies for MountainCar, instead focusing on the impacts of IV-RL. We hope this allows a smoother read and to keep the focus on our contributions.
>
> **Lack of concrete cases where IV-RL is useful**
>
> As suggested, we have *updated the conclusion* to mention concrete cases where IV-RL is particularly relevant. We underline how the performance improvement in control-based tasks could enable the application of DRL algorithms to real world problems such as robotics, where experience is expensive, and sample efficiency is thus an important challenge.
> We also added *more details about concrete applications* in the introduction.
>
> **Limitations of IV-RL**
>
> An important part of research is to evaluate the limitations of the contributions.
>
> We have identified two major limitations to IV-RL. First, since our algorithm uses ensembles, it leads to increased computation time with respect to baselines such as DQN or SAC. We have *increased the emphasis* on the additional computational power needed in section 4. We have also *improved the discussion and added results about the computational time in Appendix D*.
>
> Second, our experiments show that IV-RL does not improve the performance in exploration-based environments where there is very low information about the reward, such as MountainCar. We *identified this shortcoming in a clearer way in section 4*, and *reinforced the emphasis* on control-based tasks both in section 4 and in the conclusion.

---

> > ### Comment · Reviewer_Ncaq · 2021-11-25
> > **The paper is much improved, thank you**
> >
> > Thank you, authors, for taking the time to directly respond to the major considerations I raised in my review. I also appreciate the effort made to improve the clarity of the paper and how it now conveys information crucial to understanding the proposed IV-RL methods. I feel that the major weaknesses of the paper, as submitted, have been sufficiently addressed and can support its publication.
> >
> > Without much room to add more detail, this isn't an easy recommendation to make but I think that the paper would be better situated if the "real-world" justifications in the Introduction were more thoroughly elaborated on or discussed. The addition of the text describing these things felt a little more vague than I anticipated. I hope that the authors are afforded a little more room if the paper ultimately accepted for publication to elaborate more thoroughly in the Introduction and Conclusion sections.

---

### Official Review · Reviewer_gusa · 2021-10-31

**Correctness:** 3
**Technical Novelty And Significance:** 3
**Empirical Novelty And Significance:** 4
**Recommendation:** 8
**Confidence:** 4

**Main Review:**

Strengths

- The proposed technique is theoretically sound, and I think it makes a lot of sense to use label uncertainty (aleatoric) in Reinforcement Learning in DQN/SAC in particular, and this improves sample complexity as more information is extracted from the training process.
- I believe that it is very important to work on improving sample complexity of Reinforcement Learning, as it is the most important challenge in order to apply RL to real-world problems.
-  There is a decent improvement in sample complexity when applying BIV to DQN and SAC with no drawbacks. In two of the tested discrete environments (LunarLander and CartPole noise) there is a clear improvement, while on MountainCar there is no improvement but also the method works with same performance as the baseline. For continuous environments, in three out of four there are clear improvements in sample complexity, while ant performance is kind of the same than the best baseline and slightly down. In all cases the IV-RL methods outperform the sample complexity of the original variations without IV-RL.
- The proposed IV-RL method seems to be an improvement without drawbacks (except for additional hyper-parameters), and overall I think the idea of estimating aleatoric uncertainty in the Q-function being learned is a great idea, as this extracts additional information from the training process, the information seems to be there and it should be exploted as standard part of the Q-learning or SAC training process. This additional information is extracted using a heteroscedatic Gaussian negative log-likelihood loss which can estimate aleatoric uncertainty of the targets without supervision.
- The authors perform ablations (Figures 3 and 5) that shows the effect of different components in their proposed network, namely if using ensembles, inverse variance weights, and aleatoric uncertainty quantification through the Gaussian NLL loss.

Weaknesses

- One important detail that I think was left out the paper is the effect of the multi-task loss parameter lambda (Eq 10), if I am not mistaken, there is no information on the paper to which value this parameter is set. The supplementary mentions that it is tuned using grid search, and only the range of this parameter is provided, not the actual value that was used to obtain the results in the paper or ablations to show the effect of this parameter on performance. This is important information for the reader and reproducibility that should be included in the paper or supplementary.
- A second important issue is Eq 10, where the authors claim that the loss function they use is a standard linear combination of the BIV and LA losses, and the BIV loss is defined in Eq 1. This equation shows that the BIV loss contains another sub-loss term ( L (f(xk , θ), ỹk) ) where I assume that it is a squared error loss like the one in Eq 5, but then Eq 3 which shows the LA loss also contains a squared error weighted with the inverse variance (without the ξ term), so it would not make much sense to combine the BIV and LA losses like that directly, I think there is a missing detail here or the author's proposed loss has two squared error terms which can be inconsistent or strange. Please clarify.

Minor Points

- I think that Sections 2.2.1, 2.2.2 and 2.2.3, I think the authors narrative about deep ensembles being sampled ensembles with variance prediction is incorrect, as the original deep ensembles paper (Lakshminarayanan 2017) does not use Dropout or Masks in the ensemble network members, and there is the inconsistency that the papers the authors argue are about sampled ensembles are from 2020, while the deep ensembles paper is from 2017. Also I do not like the term "probabilistic networks" and "probabilistic ensembles", as it is not clear here what makes them probabilistic, I think better terms are for probabilistic networks are variance networks (as they output mean and variance of the targets), and plain ensembles instead of probabilistic ensembles. Also please note that deep ensembles for regression uses individual variance networks as ensemble members, so the only difference between an ensemble and a variance network is the number of members in the ensemble, where an ensemble of one network is a variance network. I think these terms will make the paper more consistent with the uncertainty quantification literature, and should also be reflected in the label for Figures 3 and 5.
- I think the paper does not mention how and why the RL environments were selected I think it is better to evaluate on more environments or at least provide a good justification for the environments that were used. Papers like SUNRISE are evaluated on many more environments (ATARI, Gym, Deepmind Control Suite) which ensures that the results and conclusions are robust.
- I think in the labels of Figure 3 and 5, it is better to relate the components of your loss function (Eq 10) and to using ensembles or single networks, instead of using the names of probabilistic networks/ensembles, this connects also with my previous comment about Sec 2.2.2 and 2.2.3.
- I would like to point the authors to another paper submitted to ICLR 2022 ( https://openreview.net/forum?id=aPOpXlnV1T ) which deals with practical issues in aleatoric uncertainty estimation using the Gaussian NLL loss for regression, which could help improve their method. In that paper the authors also propose a weighting for each loss term based on their variance.

The issues pointed above have been addressed by the authors in their newest revision.

**Summary Of The Paper:**

This paper is about uncertainty quantification in Q-Learning/Actor-Critic arquitectures. The authors deal with the problem of noise in the targets from where the Q function or value/critic function is learned, by estimating their uncertainty (aleatoric mostly) during the training process, and the authors propose to use this uncertainty with batch inverse variance weighting (BIV), in order to weight samples by inverse variance.

This idea is simple and yet quite powerful, as results show that using target uncertainty and BIV, DQN and SAC models are able to learn faster and improve their sample complexity by reaching the same average return with less number of episodes in a series of standard RL benchmarks for discrete and continuous environments.


**Summary Of The Review:**

I believe that this is a good paper, after rebuttal the issues I mentioned have been resolved. I like the idea, it is an important problem in RL, it seems to be correctly evaluated, and it will be a good contribution to the RL community as aleatoric uncertainty is generally not modelled in RL formulations.

---

> ### Author Response · Authors · 2021-11-18
> **Response to Reviewer gusa**
>
> We would like to thank you for your valuable feedback. We have made several changes to clarify the paper and address your concerns.
>
> **Role and value of the $\lambda$ hyperparameter**
>
> We agree the information about $\lambda$ is important for the reader. We have *added in part 3.1.3* that $\lambda$ is usually optimal between 5 and 10. In addition, as with every other hyperparameter, the values of $\lambda$ used for our results are available in the config file of the code we provided as supplementary material, for reproducibility purposes.
>
> We have also *added a section about $\lambda$'s influence in Appendix C*. We show experiments with different values of $\lambda$ between 1 and 10. Our results show that the exact value of $\lambda$ within this range does not strongly impact the performance of the algorithm. We also show preliminary results of the effects of the extreme values of $\lambda$. More results with a broader range of $\lambda$ values are being produced: we plan to have them ready before the end of the rebuttal period.
>
> **Equation 10 and square-error terms in the IV-RL loss**
>
> We have *modified equation 1* to include the L2 loss, as well as *equation 10* to explicitly show how the BIV and LA losses are applied.
>
> It is correct that both components have a squared error term. This is due to the design choices:
> 1) We use L2 loss for heteroscedastic regression with BIV, as this is the closest
> 2) We model the noise associated with the stochasticity of the environment and the policy as a Gaussian parametrized by its mean $\mu$ and variance $\sigma^2$. When applied to negative log likelihood loss, this gives:
> $ \mathcal{L}_{\mathrm{LA}} = -\ln(p(y)) = \frac{(y - mu)^2}{\sigma^2} + ln(\sigma^2) $, which is where the squared error term appears.
>
> IV-RL in itself is however agnostic to these choices: different loss functions for the regression or different noise models for the environment and policy stochasticity could be applied.
>
> In both components, the squared error term pushes the prediction of the Q-value towards the target, which is the basis of the Bellman equation: the two loss functions do not compete against each other on this aspect.
>
> **Terminology: variance networks and deep ensembles**
>
> Making the paper consistent with existing literature is important. We followed your suggestions and *reformulated section 2.2.3* to remove any chronological link between “sampled” and “probabilistic” ensembles. In addition, we have *changed the terms* “probabilistic network” for “variance network” across the paper. To distinguish easily between the two types of ensembles with deep neural networks, we kept “sampled ensemble” and changed “probabilistic ensembles” to “variance ensembles” (we mentioned the original term “deep ensemble” in section 2.2.3). As these changes lead to the word “variance” being used very often with different meanings (+100 times in 9 pages), sometimes several times in the same sentence, we improved readability by using “var-network” and “var-ensemble” after section 2.2.3.
>
> Finally, we *clarified the labels* in figure 3 and 5 to reflect these changes.
>
> **Choice of environments**
>
> With the resources that we had at our disposal, we decided to cover different types of environments: discrete and continuous, as well as sparse/dense rewards and control/exploration-based. We have *added a sentence explaining these choices* in section 4.1 (results).
>
> In addition, the Mujoco tasks on MBBL are a standard benchmark for robotics and control tasks, on which for example Wu et al. (2021) evaluate their results. We have also *added results for the SlimHumanoid environment* in section 4.2.
>
> Finally, the significance of DRL experiments is known to be an issue (Henderson et al., 2019). We found it important to follow a thorough experimental methodology. As described in Appendix B, we performed rigorous hyperparameter tuning for all baselines over 5 seeds, and then repeated the 4 best configurations over 25 seeds for final results. This procedure is computationally heavy, and for the same amount of resources, it reduces the variety of environments we can experiment on. However, it adds robustness and statistical significance to the results.
>
> **Clarity of labels**
>
> We have *clarified the text explaining the ablation study* as well as the labels for figures 3 and 5.
>
> **Paper suggestion: On the Pitfalls of Heteroscedastic Uncertainty Estimation with Probabilistic Neural Networks**
>
> Thank you for pointing us towards this interesting paper. It is indeed relevant to our work: the pitfalls presented in the paper could very well happen in our pipeline with the use of Gaussian negative log-likelihood loss, and be responsible for overestimated variance estimation as well as bad policy evaluation. We will consider the use of the proposed method in our future work on variance estimation in deep reinforcement learning.

---

> > ### Author Response · Authors · 2021-11-21
> > **Notification: new results for the effect of $\lambda$**
> >
> > Dear Reviewer,
> >
> > We would like to notify you that we have now added the full results regarding the role of $\lambda$ in IV-RL, in appendix C.
> >
> > Thank you,
> >
> > The authors

---

> > ### Comment · Reviewer_gusa · 2021-11-22
> > **Good rebuttal, addresses my issues**
> >
> > Dear Authors, thank you for your detailed rebuttal, I think your new results and changes do address the issues I mentioned in my review and I will be updating my score accordingly.
> >
> > The only issue that I think remains is minor, the double squared errors in Eq 10, maybe you could see in the future a way to merge both squared errors in order for the reader not find these surprising, but seems this is not an issue in practice.

---

### Official Review · Reviewer_9Z1v · 2021-11-02

**Correctness:** 4
**Technical Novelty And Significance:** 3
**Empirical Novelty And Significance:** 2
**Recommendation:** 8
**Confidence:** 4

**Main Review:**

**Clarity.**

The paper is well-written and fairly clear in its justification, background, and empirical results. However, the lack of clear math and pseudocode applying the method to the RL setting makes it unclear as to what exactly the method is and how to implement it. For the next iteration of the paper, this information is necessary. Including only equations for supervised learning (i.e., equations 1, 3, 10) for one of the main contributions of the paper is insufficient in an RL paper. Minor: The math in section 3.1.2 seems clear and correct, although should there be an $s,a$ in the RHS conditioning of equation 8?


**Novelty and significance.**

The main components of the proposed method (i.e., down-weighting by variance and using probabilistic ensembles) are well-known to the community, as cited already by this paper (e.g., Lee et al, 2020, Wu et al, 2021, and Lakshminarayanan et al, 2017). The particular proposed method for down-weighting is taken directly from supervised learning and applied to RL as a method for the above combination. Thus the novelty is just the combination of these, which is insufficient in my view to merit acceptance. The analysis of the sources of uncertainty (sec 3.1.2) is perhaps novel (I did not do a thorough literature review given the other issues with the paper) but insufficient to merit acceptance.

Despite the lack of novelty, if the theoretical or empirical results were strong, then acceptance could be warranted. However, the empirical evidence here is limited and thus does not tip the balance. The environments are all very similar and none are particularly challenging, so it’s unclear the results will hold for a broader range of environments (for example, only 1 of the 3 discrete environments actually improve due to BIV). Further, if the argument is that the normalization used in BIV is crucial relative to UWAC, then I’d like to see this explicitly. I’d similarly like to see a more detailed set of experiments comparing the weighting schemes of UWAC, SUNRISE, and BIV (with all else identical) to know how they compare.

**Question:** how does variance down-weighting interact with the benefits of optimism in the face of uncertainty in RL? Down-weighting by high variance elements of the batch could be interpreted as a form of pessimism, which would run counter to this.


**Summary Of The Paper:**

The paper presents inverse-variance RL, which combines an ensemble of probabilistic neural networks with batch inverse-variance weighting. The former provides a method for uncertainty estimation and the latter provides a method for incorporating that uncertainty, by multiplying the loss of each entry in each minibatch by the (normalized) inverse of its variance. Experiments on simple discrete and continuous control environments from the OpenAI Gym show that the resulting method, when added to DQN and SAC outperforms these baselines and SUNRISE.

**Summary Of The Review:**

Overall, this paper lacks sufficient novelty for acceptance. It recombines a set of existing techniques in a slightly different way than has been done before and shows that this seems to improve performance on the small set of fairly similar domains presented. However, for such a minor change from what has come before, it lacks sufficiently strong theoretical or empirical evidence for acceptance.


----------

After rebuttal, the majority of my concerns have been addressed by a comprehensive update to the paper. To reflect this, I have updated my scores to accept.

---

> ### Author Response · Authors · 2021-11-18
> **Response to Reviewer 9Z1v (1/2)**
>
> Thank you for your detailed review of our work. We would like to underline several changes that we have made to improve the paper as a result of your comments.
>
> **Lack of clear math and pseudocode with respect to the RL implementation**
>
> We have addressed the issue of lack of pseudocode and clear math by *adding detailed algorithms and extensive descriptions of IV-DQN, IV-SAC, as well as the baselines and the ablated methods, in appendix A*. Additionally, we have *clarified the loss function in equation 10* to make it more explicit. We are confident that this improves the clarity of the paper. Thank you for identifying this shortcoming.
>
> **Section 3.1.2: missing s,a in the conditioning**
>
> We have *corrected* this. Thank you!
>
> **Novelty and significance**
>
> We would like to re-highlight some of the novel aspects of this paper:
> - By identifying the connections between TD learning and regression on heteroskedastic noisy labels, we provide a theoretical motivation for the use of probabilistic (variance) ensembles (Lakshminarayanan et al, 2017). To the best of our knowledge, this is the first work using variance ensembles to improve the supervision process in DRL. Both Lee et al, 2020 and Wu et al, 2021 instead framed the problem as out-of-distribution detection, and thus focused on epistemic uncertainty using sampled ensembles or MC-Dropout. By the combined effect of a better predictive variance estimation and the down-weighting happening in negative non-likelihood loss, our approach leads to significantly better performance, without any drawbacks.
> - Using probabilistic ensembles involves training the variance prediction, which brings some challenges, such as early variance underestimation as well as miscalibrated uncertainty prediction. We show that using the normalized BIV framework together with minimal effective batch size addresses these issues better than the weighting schemes used by Lee et al. 2020 or Wu et al 2021.
> - Our method is the result of many experiments built upon a strong theoretical motivation. It leads to novel insights about the use of uncertainty in DRL, and to significantly improved performance. We believe there is an empirical value to the community in summarizing these findings.
>
> **Range of environments**
>
> With the limited resources that we had at our disposal, we decided to cover different types of environments: discrete and continuous, as well as sparse/dense rewards and control/exploration-based. We have *added a sentence underlining this diversity* in section 4.1 (results).
>
> In discrete environments, the improvement is significant in 2 out of 3 environments: LunarLander and CartpoleNoise. As can be expected, IVRL does not improve the performance in exploration-heavy environments such as MountainCar.
>
> In addition, we ran the continuous tasks on MBBL with Mujoco. This is a standard benchmark for robotics and control tasks, on which for example Wu et al. (2021) evaluate their results. We have also *added additional results for the SlimHumanoid environment* in section 4.2.
>
> Finally, the significance of DRL experiments is known to be an issue (Henderson et al., 2019). We found it important to follow a thorough experimental methodology. As described in Appendix B, we performed rigorous hyperparameter tuning for all baselines over 5 seeds, and then repeated the 4 best configurations over 25 seeds for final results. This procedure is computationally heavy, and for the same amount of resources, it reduces the variety of environments we can experiment on. However, it adds robustness and statistical significance to the results.
>
> **Difference with UWAC**
>
> As seen in figure 5, UWAC and Sunrise weights show inferior performance to IV-SAC on HalfCheetah and Walker, when applied to variance ensembles (probabilistic ensembles), because of both the normalization and MEBS elements. We have *added new results and a detailed discussion about the difference between the weighting schemes* in Appendix F.
>
> Note that the original UWAC and Sunrise methods do not use variance ensembles, which is a critical part of IV-RL. With sampled ensembles, UWAC would be very similar to BIV + BootstrapDQN or BIV+EnsembleSAC, while the Sunrise results are given in figures 2 and 4 and are clearly suboptimal.

---

> > ### Author Response · Authors · 2021-11-18
> > **Response to Reviewer 9Z1v (2/2)**
> >
> > **Interaction with Optimism in the Face of Uncertainty**
> >
> > We would like to thank the reviewer for raising this question. We have had this discussion while working on this project, and are happy you mentioned it. We *added a short discussion about this topic* in Appendix H, that we summarize here.
> >
> > It’s important to distinguish the uncertainty of the value estimation at state-action pair $(s,a)$ and the uncertainty at $(s’, a’)$. The former is used in optimism in the face of uncertainty as an exploration strategy to choose the next action: we encourage the agent to get more experience in the areas of the state space it does not know well. The latter is considered by IV-RL when computing the target’s variance: we encourage the agent to learn more from trustworthy signals when updating the value function. These are different values, and at first view, OFU and IV-RL do not interact.
> >
> > However, the sequential nature of experience in RL induces a correlation in the information the agent has about the value of $(s,a)$ and the value of $(s’,a’)$. Additionally, in the context of function approximation, there may be a relation between the uncertainties of $(s,a)$ and $(s’,a’)$ as these two pairs may be close. These points can affect the orthogonality of OFU and IV-RL, which may interact. In this case, the use of minimal effective batch size (MEBS) to control the discrimination due to IV-RL allows to balance both methods.
> >
> > We show in this paper that, empirically, both strategies of OFU and IV-RL are compatible: in IV-SAC, a UCB exploration bonus is used with success. However, we believe this interaction could be a very interesting topic of future research.

---

> > ### Comment · Reviewer_9Z1v · 2021-11-25
> > **Update after revisions**
> >
> > Thank you for the comprehensive response and updates to the paper. The addition of proper descriptions of the math and algorithms, and the careful ablations and discussion with respect to prior work has done much to address my concerns of novelty. While the proposed approach does share much with existing work, the additional experiments show that the differences do seem to have significant positive effects and thus the work will be of interest to the community. As such, I've updated my score to accept.
> >
> > It would still be great to see additional experiments on other types of environments (namely, RL from pixels instead of just state) but I do appreciate the thoroughness of the hyperparameter optimization done for all methods (baseline and proposed).
> >
> > One other piece of related work that also explores the respective roles of variance networks and ensembles of them is "Selective Dyna-Style Planning Under Limited Model Capacity" by Abbas, Sokota, Talvitie, and White, which is probably worth mentioning, as it supports some of the conclusions found here (namely the results in appendices E and F).

---

### Official Review · Reviewer_HGRr · 2021-11-03

**Correctness:** 4
**Technical Novelty And Significance:** 3
**Empirical Novelty And Significance:** 4
**Recommendation:** 10
**Confidence:** 5

**Main Review:**

This paper is exceptionally well written and a pleasure to read. I literally found no weaknesses and strongly recommend acceptance.

Strengths:

1. It is an important research topic and of very high interest to the community.
2. The approach is very well embedded in the current literature.
3. The paper is very easy to follow. It is theoretically sound and for all critical aspects there are intuitions and explanations.
4. The evaluation is done very thoroughly (results over 25(!) seeds, hyperparameters are provided).
5. The benefits of IV-RL are shown in both the discrete and continuous setting.

Weaknesses:
None

Remark:
I would be interested in a more extensive discussion of policy degeneration in the LunarLander environment.

**Summary Of The Paper:**

In the paper "Sample Efficient Deep Reinforcement Learning via Uncertainty Estimation", a group of algorithms called "Inverse Variance Reinforcement Learning" is proposed that leverages different estimations of uncertainty to guide the loss of value-functions (and actors in actor-critic settings). IV-RL is evaluated for IV-DQN and IV-SAC in several benchmarks.

**Summary Of The Review:**

I literally found no weaknesses and strongly recommend acceptance.

---

> ### Author Response · Authors · 2021-11-18
> **Response to Reviewer HGRr**
>
> Thank you for your positive feedback and for recognizing our contributions.
>
> **To answer your remark about the policy degeneration in Lunar Lander:**
>
> We have examined these results in detail after submission, as we also thought this needed to be better understood. We found out that, in this particular case, we had not used BootstrapDQN as an exploration strategy, but epsilon-greedy instead. This was thus not a fair comparison, as the baselines were using BootstrapDQN. We have produced new results where BootstrapDQN is used with IV-DQN. The degeneration is prevented and the performance is better than previously reported. Being an approximation of Thompson sampling, BootstrapDQN is indeed more robust and stable than e-greedy exploration.

---

### Author Response · Authors · 2021-11-18
**General response to the reviewers**

We would like to thank all the reviewers for the detailed reviews and valuable feedback.
We are glad that you recognized the novelty in our approach to TD learning as heteroscedastic regression, the overall clarity of the paper, the relevance of the sample efficiency problem in DRL, and the robustness and significance of our results.

The main concern that was identified in the reviews was the lack of clarity of the method. We have made several modifications in this respect, such as the addition of pseudo-code and detailed descriptions, as well as a more explicit loss function.
Several additional issues were raised, which we have addressed in the individual answers. Every modification to the manuscript is written in blue.

Should there be any additional comments, we are available to address them before the end of the discussion period.

Thank you once again,

The authors

---

### Decision · Program_Chairs · 2022-01-20

**Decision:**

Accept (Spotlight)

**Comment:**

The reviewers unanimously appreciated the clarity of the work as well as the framing of the proposed method. Congratulations.